# Deletion of Calsyntenin-3, an atypical cadherin, suppresses inhibitory synapses but increases excitatory parallel-fiber synapses in cerebellum

Zhihui Liu[1,2]\*, Man Jiang[1†], Kif Liakath-Ali[1], Alessandra Sclip[1,2], Jaewon Ko[3], Roger Shen Zhang[1], Thomas C Südhof[1,2]\*

[1]Department of Molecular and Cellular Physiology, Stanford University, Stanford, United States; [2]Howard Hughes Medical Institute, Stanford University, Stanford, United States; [3]Department of Brain Sciences, Daegu Gyeongbuk Institute of Science and Technology, Daegu, Republic of Korea

**Abstract** Cadherins contribute to the organization of nearly all tissues, but the functions of several evolutionarily conserved cadherins, including those of calsyntenins, remain enigmatic. Puzzlingly, two distinct, non-overlapping functions for calsyntenins were proposed: As postsynaptic neurexin ligands in synapse formation, or as presynaptic kinesin adaptors in vesicular transport. Here, we show that, surprisingly, acute CRISPR-mediated deletion of calsyntenin-3 in mouse cerebellum in vivo causes a large decrease in inhibitory synapse, but a robust increase in excitatory parallel-fiber synapses in Purkinje cells. As a result, inhibitory synaptic transmission was suppressed, whereas parallel-fiber synaptic transmission was enhanced in Purkinje cells by the calsyntenin-3 deletion. No changes in the dendritic architecture of Purkinje cells or in climbing-fiber synapses were detected. Sparse selective deletion of calsyntenin-3 only in Purkinje cells recapitulated the synaptic phenotype, indicating that calsyntenin-3 acts by a cell-autonomous postsynaptic mechanism in cerebellum. Thus, by inhibiting formation of excitatory parallel-fiber synapses and promoting formation of inhibitory synapses in the same neuron, calsyntenin-3 functions as a postsynaptic adhesion molecule that regulates the excitatory/inhibitory balance in Purkinje cells.

\*For correspondence:
zhihuil@stanford.edu (ZL);
tcs1@stanford.edu (TCS)

Present address: [†]Department of Physiology, School of Basic Medicine and Tongji Medical College, Huazhong University of Science and Technology, Wuhan, China

Competing interest: The authors declare that no competing interests exist.

## Editor's evaluation

This study investigated the role of calsyntenin-3, an atypical cadherin, in controlling synaptic inputs to cerebellar Purkinje cells. It provides compelling evidence that elimination of calsyntenin-3 from cells in the cerebellar cortex alters the E/I balance for Purkinje cells, although Purkinje cell-specific manipulations were used in only some of the experiments. The results indicated that calsyntenin-3 increases the strength of excitatory parallel fiber inputs and decreases the strength of inhibitory inputs.

## Introduction

Synapses mediate information transfer between neurons in brain and process the information during transfer. In processing information, synapses are dynamic: Synapses are continuously eliminated and newly formed throughout life, and are additionally restructured by various forms of synaptic plasticity during their life cycle (*Attardo et al., 2015*; *Pfeiffer et al., 2018*). Synapse formation, elimination, and remodeling are thought to be organized by synaptic adhesion molecules (SAMs) (*Südhof, 2021*).

Many candidate SAMs have been described, which enact distinct facets of synapse formation and collaborate with each other in establishing synapses and specifying their properties. However, few of the many candidate SAMs described appear to contribute to the initial establishment of synapses. At present, only two families of adhesion-GPCRs, latrophilins and BAIs, are known to have a major impact on synapse numbers when tested using rigorous genetic approaches (*Anderson et al., 2017*; *Bolliger et al., 2011*, *Kakegawa et al., 2015*; *Sando et al., 2019*; *Sando and Südhof, 2021*; *Sigoillot et al., 2015*; *Wang et al., 2020*). In contrast, the majority of SAMs that have been studied, most prominently neurexins and LAR-type receptor phosphotyrosine phosphatases, appear to be dispensable for establishing synaptic connections. Instead, these SAMs are essential for conferring onto existing synapses specific properties that differ between various types of synapses in a neural circuit (*Chen et al., 2017*; *Emperador-Melero et al., 2021*; *Fukai and Yoshida, 2021*, *Missler et al., 2003*; *Sclip and Südhof, 2020*).

Calsyntenins (a.k.a. alcadeins) are atypical cadherins that are encoded by three genes in mammals (*Clstn1-3* in mice) and a single gene in *Drosophila*, *C. elegans*, and other invertebrates (*Araki et al., 2003*; *Hintsch et al., 2002*; *Ohno et al., 2014*; *Vogt et al., 2001*). Calsyntenins are type I membrane proteins containing two N-terminal cadherin domains followed by a single LNS-domain (also referred to as LG-domain), a transmembrane region, and a short cytoplasmic tail. Calsyntenins are primarily expressed in neurons, although a variant of *Clstn3* with a different non-cadherin extracellular sequence is present in adipocytes (referred to as Clstn3β; *Zeng et al., 2019*). The evolutionary conservation, cadherin domains, and neuron-specific expression of calsyntenins has spawned multitudinous studies on their biological significance, but no clear picture of their fundamental activities has emerged.

Two different views of calsyntenin functions were proposed. The first view posits that calsyntenins are postsynaptic adhesion molecules that bind to presynaptic neurexins to mediate both excitatory and inhibitory synapse formation (*Vogt et al., 2001*; *Pettem et al., 2013*; *Um et al., 2014*; *Kim et al., 2020*). The second view, in contrast, suggests that calsyntenins are presynaptic adaptor proteins that mediate kinesin function in axonal transport (*Araki et al., 2003*). Extensive and sometimes compelling evidence supports both views.

The first view positing a role for calsyntenins as postsynaptic adhesion molecules was spawned by the localization of all calsyntenins by immunoelectron microscopy to postsynaptic densities of excitatory synapses in cortex and cerebellum (*Hintsch et al., 2002*; *Vogt et al., 2001*). In further support of this view, calsyntenins were shown to induce presynaptic specializations in heterologous synapse formation assays when expressed in non-neuronal cells (*Pettem et al., 2013*). Most importantly, knockout (KO) mice of all three calsyntenins exhibited synaptic impairments (*Kim et al., 2020*; *Lipina et al., 2016*; *Pettem et al., 2013*; *Ster et al., 2014*). Careful analyses revealed that *Clstn3* KO mice display a 20–30% decrease in excitatory synapse density in the CA1 region of the hippocampus (*Kim et al., 2020*; *Pettem et al., 2013*). In addition, *Pettem et al., 2013* observed a similar decrease in inhibitory synapse density in the CA1 region, although *Kim et al., 2020* failed to detect such a decrease. Moreover, *Pettem et al., 2013* reported a 30–40% decrease in mEPSC and mIPSC frequency, but unexpectedly found no change in excitatory synaptic strength as measured by input/output curves. Similar analyses showed that the *Clstn2* KO also decreased the inhibitory synapse density in the hippocampus by approximately 10–20% (*Lipina et al., 2016*), whereas the *Clstn1* KO modestly impaired excitatory synapses in juvenile but not in adult mice (*Ster et al., 2014*). Viewed together, these data suggested a postsynaptic role for calsyntenins in the hippocampus, although the modest effect sizes of the calsyntenin KO phenotypes were puzzling.

As postsynaptic adhesion molecules, calsyntenins were proposed to function by binding to presynaptic neurexins. However, distinct mutually exclusive mechanisms of neurexin binding were described. *Pettem et al., 2013* and (*Lu et al., 2014*) showed that the LNS domain of calsyntenins binds to an N-terminal sequence of α-neurexins that is not shared by β-neurexins. *Kim et al., 2020*, in contrast, demonstrated that the cadherin domains of calsyntenins bind to the 6th LNS domain of neurexins that is shared by α- and β-neurexins. Adding to this puzzle, *Um et al., 2014* did not detect any direct binding of calsyntenins to neurexins, and no study has reconstituted a stable calsyntenin-neurexin complex. Viewed together, these data established a postsynaptic organizing function of calsyntenins in synapse formation by an unknown mechanism that may involve neurexins.

The alternative view, namely that calsyntenins function as presynaptic kinesin-adaptor proteins that facilitate vesicular transport, is also based on extensive evidence. This view was motivated by the

localization of calsyntenins to transport vesicles containing APP (*Araki et al., 2007*; *Konecna et al., 2006*; *Vagnoni et al., 2012*). A cytoplasmic sequence of calsyntenins binds to kinesins (*Konecna et al., 2006*), and at least *Clstn1* found in vesicles containing kinesin (*Ludwig et al., 2009*). Moreover, carefully controlled immunoprecipitations showed that calsyntenins are present in a molecular complex with presynaptic GABA$_B$-receptors and APP (*Dinamarca et al., 2019*; *Schwenk et al., 2016*). However, *Clstn1* KO mice exhibited only modest changes in APP transport and in the proteolytic processing of APP into Aβ peptides, and *Clstn2* KO mice displayed no changes in these parameters (*Gotoh et al., 2020*). Furthermore, the *Clstn1* KO simultaneously increased the levels of the C-terminal cleavage fragment (CTF) of APP and of Aβ peptides without changing APP levels, making it difficult to understand how a decreased APP cleavage causing increased CTF levels could also elevate Aβ levels (*Gotoh et al., 2020*). As a result, the kinesin binding, APP interaction, and GABA$_B$-receptor complex formation by calsyntenins are well established, but it is not yet clear how these activities converge on a function for calsyntenins in axonal transport of APP.

The two divergent views of the function of calsyntenins are each well supported but difficult to reconcile. Given the potential importance of calsyntenins, we here pursued an alternative approach to study their functions. We aimed to identify neurons that express predominantly one calsyntenin isoform in order to avoid potential redundancy, and then examined the function of that calsyntenin isoform using acute genetic ablations and synapse-specific electrophysiological analyses. Our results reveal that cerebellar Purkinje cells express only *Clstn3* at high levels. Using CRISPR/Cas9-mediated deletions, we unexpectedly found that deletion of *Clstn3* in Purkinje cells upregulated excitatory parallel-fiber synapses and had no effect on excitatory climbing-fiber synapses, but suppressed inhibitory basket- and stellate-cell synapses. These results demonstrate that in Purkinje cells, *Clstn3* unequivocally functions as a postsynaptic synaptic adhesion molecule, but that *Clstn3* in these neurons surprisingly differentially controls the relative abundance of different types of synapses instead of simply supporting formation of subsets of synapses.

## Results

### *Clstn3* is the predominant calsyntenin isoform of cerebellar Purkinje cells

To analyze physiologically relevant functions of calsyntenins, we aimed to identify a type of neuron that expresses a particular calsyntenin isoform at much higher levels than others. This was necessary to avoid the potential for functional redundancy among multiple calsyntenins that may have occluded phenotypes in previous KO analyses (*Kim et al., 2020*; *Lipina et al., 2016*; *Ster et al., 2014*; *Pettem et al., 2013*). Since previous studies on calsyntenin functions were performed in the hippocampus, we examined calsyntenin expression in the hippocampus using single-molecule in situ hybridizations. We confirmed that all three calsyntenins were expressed in the CA1 and CA3 regions and the dentate gyrus (*Figure 1A*). In the CA1 region, *Clstn1* and *Clstn3* levels were highest; in the CA3 region, all three calsyntenins were similarly abundant; and in the dentate gyrus, *Clstn1* and *Clstn2* were most strongly present (*Figure 1A*). These results suggest that most hippocampal neurons co-express multiple calsyntenin isoforms, which may account for the modest phenotypes observed with *Clstn1*, *Clstn2*, and *Clstn3* KO mice (*Kim et al., 2020*; *Lipina et al., 2016*; *Pettem et al., 2013*; *Ster et al., 2014*).

We next examined the cerebellum because single-cell RNA transcriptome databases suggested that in the cerebellum, granule cells express primarily *Clstn1*, while Purkinje cells selectively express *Clstn3*, suggesting that the expression of calsyntenin isoforms is more segregated in cerebellum than in the hippocampus (*Peng et al., 2019*; *Saunders et al., 2018*; *Schaum et al., 2018*; *Tasic et al., 2018*; *Zeisel et al., 2018*). Indeed, quantifications of publicly available single-cell RNAseq data indicated that Purkinje cells in the cerebellum express more than 15-fold higher levels of *Clstn3* than *Clstn1* or *Clstn2*, whereas other cell types in the cerebellum primarily express *Clstn1* (*Figure 1—figure supplement 1A*). Single-molecule in situ hybridization of cerebellar sections corroborated that Purkinje cells express almost exclusively *Clstn3*, although low amounts of *Clstn2* were also detectable, whereas granule cells express *Clstn1* (*Figure 1B and C*). The differential labeling signal for the calsyntenins in the cerebellum was not due to differences in probe efficiency because in the hippocampus, the same probes under the same conditions produced equally strong signals (*Figure 1A*).

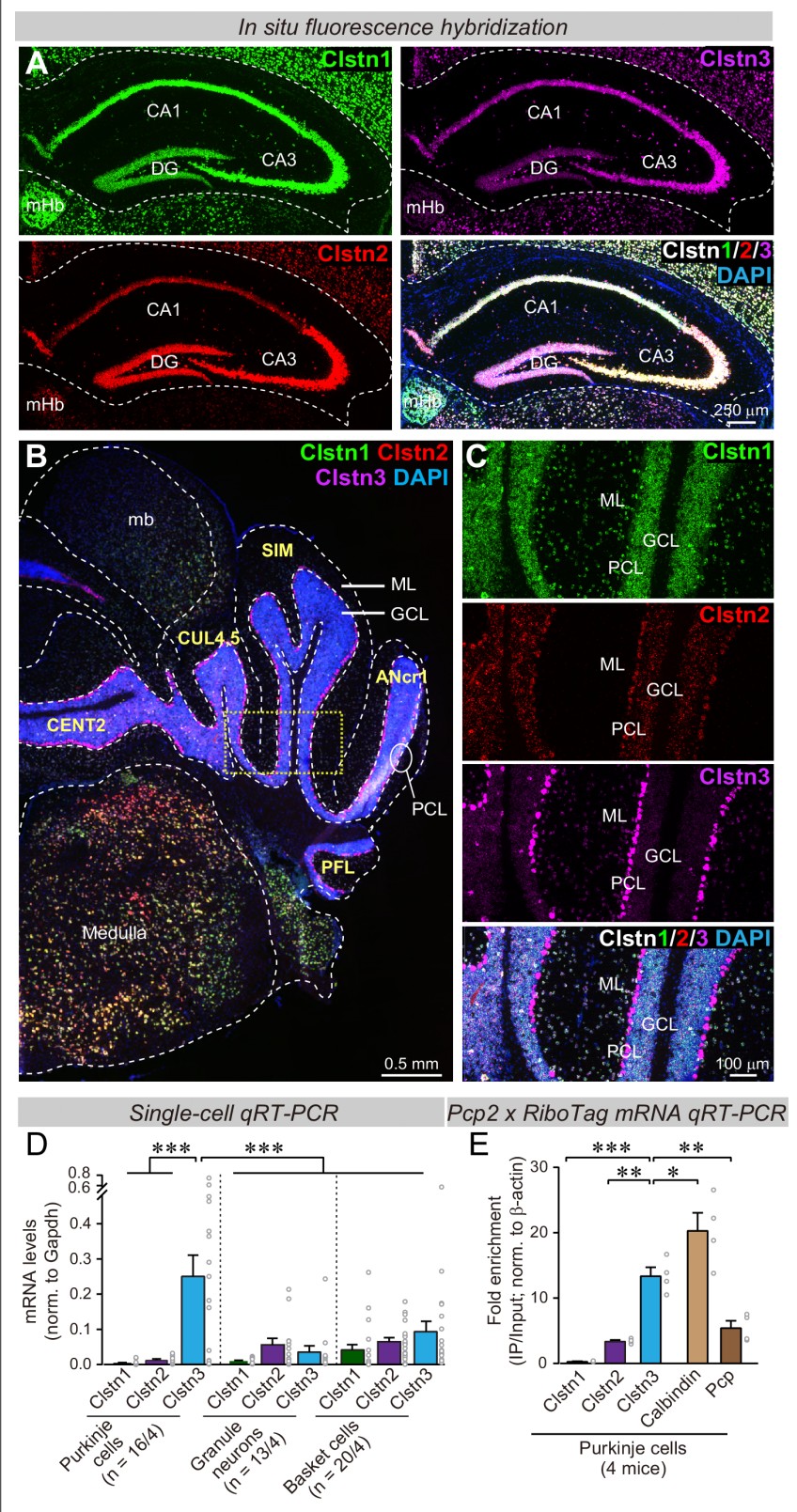

**Figure 1.** *Clstn1*, *Clstn2*, and *Clstn3* are co-expressed in overlapping neuronal populations in the mouse hippocampus, but are largely expressed in separate neuronal populations in the mouse cerebellum. (**A**) Single-molecule in situ fluorescent hybridization (RNAscope) reveals that *Clstn1*, *Clstn2* and *Clstn3* exhibit largely overlapping expression patterns in the dorsal hippocampus. Representative images show sections from a mouse

*Figure 1 continued*

at P30 labeled with probes to *Clstn1* (green), *Clstn2* (red), and *Clstn3* (magenta) and with DAPI (blue) as indicated (DG, dentate gyrus; CA1 and CA3, CA1- and CA3-regions of the hippocampus proper; mHb, medial habenula). (**B & C**) Different from the hippocampus, *Clstn1* (green), *Clstn2* (red), and *Clstn3* (magenta) exhibit largely distinct, non-overlapping expression patterns in the cerebellum as visualized by single-molecule in situ hybridization (B, overview; C, expanded views of the area boxed in B; Mb, midbrain; ML, molecular layer; PCL, Purkinje cell layer; GCL, granule cell layer). Scale bars apply to all images in a set. (**D**) Single-cell qRT-PCR demonstrates that Purkinje cells uniquely express high levels of *Clstn3* mRNAs. The cytosol of singe Purkinje, granule and basket cells in acute cerebellar slices from wild-type mice (n = 4 mice; number of cells are indicated in the graph) was aspirated via a glass pipette and subjected to qRT-PCR with primers validated in *Figure 1—figure supplement 1B*. mRNA levels were normalized to those of Gapdh using threshold cycle values (Ct). (**E**) Analyses of ribosome-associated mRNAs isolated from Purkinje cells confirms enrichment of *Clstn3* expression in Purkinje cells. Ribosome-associated mRNAs were immunoprecipitated using HA-antibodies from the cerebellum of RiboTag mice that had been crossed with Pcp2-Cre mice. mRNA levels of *Clstn1*, *Clstn2*, *Clstn3*, Calbindin, and Pcp (Purkinje cell protein-2) were measured using qRT-PCR and normalized to the internal control of β-actin using threshold cycle values (Ct). Samples were from 4 mice. Data in D and E are means ± SEM. Statistical analyses were performed using one-way ANOVA and post-hoc Tukey tests for multiple comparisons. For D, $F_{(8, 138)}$ = 8.786, p < 0.000. ***denotes p < 0.001. For E, $F_{(4,15)}$=33.065, p < 0.000. *denotes p < 0.05, **denotes p < 0.01, ***denotes p < 0.001.

The online version of this article includes the following figure supplement(s) for figure 1:

**Figure supplement 1.** Unbiased single-cell RNAseq analyses demonstrate that most *Clstn3* in the cerebellum is expressed in Purkinje cells, which in turn express little *Clstn1* or *Clstn2*, whereas other cells in cerebellum primarily express other calsyntenins (**A**), and validation of the qRT-PCR primers using standard curves to ensure that the qRT-PCR measurements are reliable (B-E, *Clstn1*, *Clstn2*, *Clstn3*, and Gapdh).

To further confirm the cell-type specificity of *Clstn3* expression in cerebellum, we patched Purkinje cells, granule cells, and basket cells in acute cerebellar slices, and performed quantitative single-cell RT-PCR measurements on the cytosol from these cells. For these experiments, we specifically validated the PCR primers (*Figure 1—figure supplement 1B-E*). These single-cell qRT-PCR measurements confirmed that Purkinje cells exclusively express high levels of *Clstn3* ( > 20-fold more than *Clstn1* or *Clstn2*), whereas other cerebellar neurons do not (*Figure 1D*). Finally, we performed qRT-PCR measurements on Purkinje cell mRNAs isolated using L7-Cre-dependent ribotag isolation (*Sanz et al., 2009*), and replicated the highly selective expression of *Clstn3* in Purkinje cells (*Figure 1E*). Thus, we demonstrated using four independent approaches (analysis of unbiased single-cell RNAseq data, in situ hybridizations, qRT-PCRs of single cells, and qRT-PCRs of bulk RNA isolated via RiboTag techniques from Purkinje cells) that cerebellar Purkinje cells selectively express high levels of *Clstn3*, whereas other cerebellar neurons express low to undetectable levels. Since Purkinje cells represent an excellent experimental system for synaptic physiology, we therefore decided to focus on the function of *Clstn3* in these neurons.

## In vivo CRISPR efficiently deletes *Clstn3* in Purkinje cells of the cerebellar cortex

Advances in CRISPR-mediated genetic manipulations suggest that it is possible to dissect a gene's function using acute CRISPR-mediated deletions in vivo (*Incontro et al., 2014*). Upon testing multiple single-guide RNAs (sgRNAs) for the *Clstn3* gene, we identified two sgRNAs targeting exons 2 and 3 of the *Clstn3* gene that potently suppressed *Clstn3* expression in the cerebellum in vivo (*Figure 2A–E*). AAVs (DJ-serotype) encoding the sgRNAs and tdTomato (as an expression marker) were stereotactically injected at P21 into the cerebellum of mice that constitutively express *sp*Cas9 (*Platt et al., 2014*), and mice were analyzed at ~P50 by quantitative RT-PCR, immunoblotting, and imaging (*Figure 2C–E*). The CRISPR-mediated *Clstn3* KO was highly effective in vivo, decreasing *Clstn3* mRNA levels by >60% (*Figure 2D and E*, *Figure 2—figure supplement 1A*), and *Clstn3* protein levels by ~80% (*Figure 2E*). The AAV-DJ serotype we used primarily infected Purkinje cells as evidenced by the selective presence of the tdTomato signal in Purkinje cells, with very little expression in granule cells and some expression in basket and stellate cells (*Figure 2F*). Although the tdTomato signal in the infected cerebellar cortex decreased with the distance from the injection site as expected since tdTomato has to diffuse along the extended dendritic tree of Purkinje cells (*Figure 2F and G*), magnifications of the distal cortical areas clearly demonstrated tdTomato expression throughout the cortex (*Figure 2—figure supplement 2*).

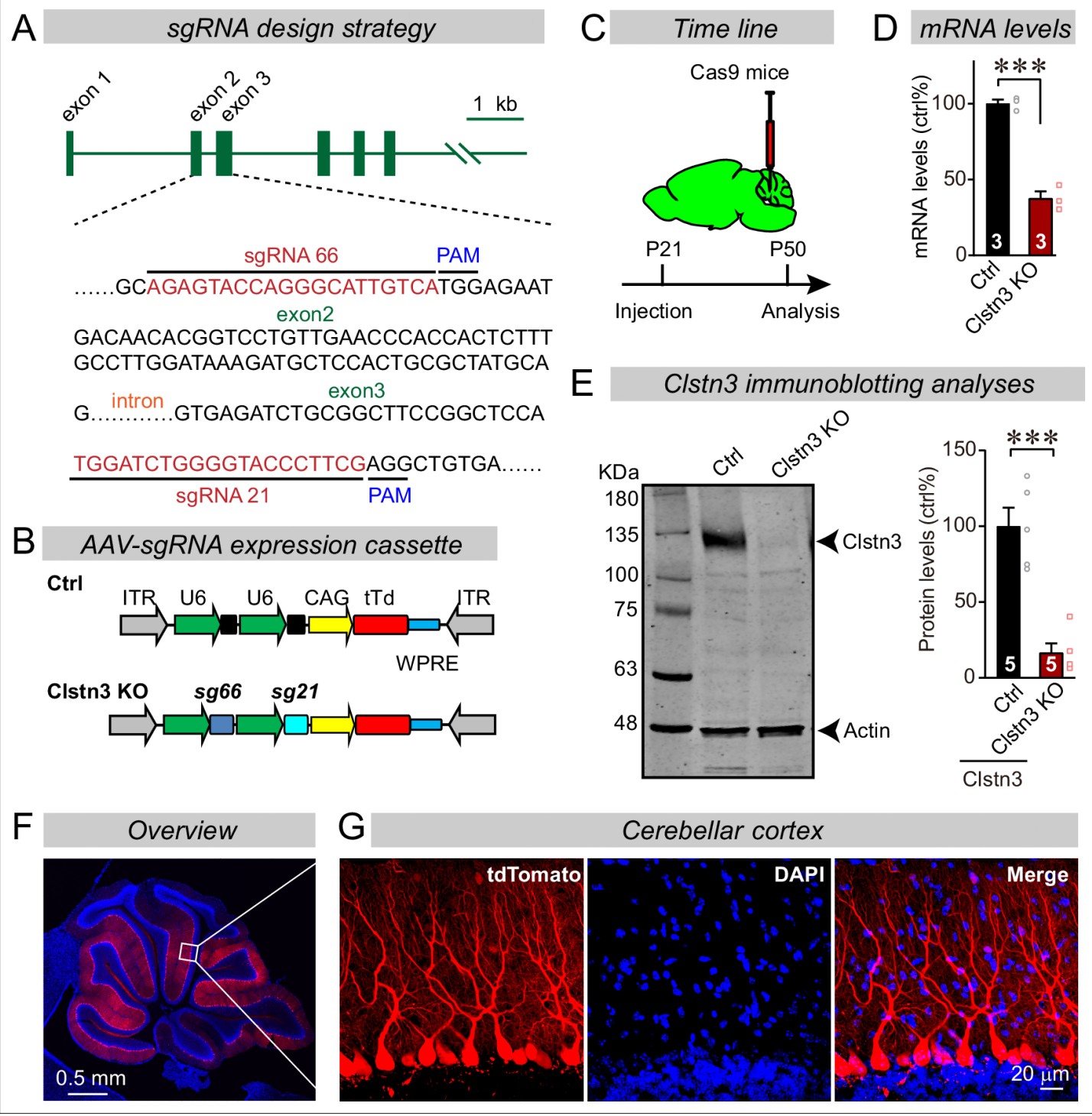

**Figure 2.** CRISPR/Cas9 manipulations enable rapid and highly efficient in vivo deletions of *Clstn3* in Purkinje cells. (**A**) Schematic of the sgRNA design strategy. Both sgRNAs target the positive strand of DNA, with sg66 targeting exon2, and sg21 targeting exon3. (**B**) Schematic of the AAV-DJ expression cassette in which sgRNAs and tdTomato (tdT) synthesis are driven by U6 and CAG promoters, respectively. Control mice were infected with AAVs that lacked sgRNAs but were otherwise identical. (**C**) Experimental strategy for CRISPR-mediated acute *Clstn3* deletions in the cerebellum. AAVs expressing the sgRNAs and tdTomato were stereotactically injected into the cerebellum of constitutively expressing Cas9 mice at P21, and mice were analyzed after P50. (**D**) Quantitative RT-PCR shows that the CRISPR-mediated *Clstn3* deletion severely suppresses *Clstn3* mRNA levels in the total cerebellum. Relative gene expression levels were first normalized to GAPDH using threshold cycle (CT) values, and then normalized to control. (**E**) Immunoblotting analyses confirm that the CRISPR-mediated deletion greatly suppresses *Clstn3* protein levels in the overall cerebellum (left, representative immunoblot; right, summary graph of quantifications using fluorescently labeled secondary antibodies). (**F & G**) Representative images of a sagittal cerebellar

*Figure 2 continued on next page*

*Figure 2 continued*

section from a mouse that was stereotactically infected with AAVs as described in **C** (F, overview of the cerebellum; G, cerebellar cortex; red = AAV-encoded tdTomato; blue, DAPI). Note that AAVs infect all Purkinje cells but few granule cells or inhibitory neurons. Data in panels D and E are means ± SEM. Statistical analyses were performed using double-tailed unpaired t-test for D and E (***p < 0.001). Numbers of animals for each experiment are indicated in the graphs.

The online version of this article includes the following figure supplement(s) for figure 2:

**Figure supplement 1.** Predicted genome editing patterns by the sgRNAs used for the *Clstn3* KO in the current study and analysis of potential off-target effects of the sgRNAs using genomic sequencing of targeted *Clstn3* KO cerebellum.

**Figure supplement 2.** The more distal cortex of the cerebellum is also infected by the AAVs that were stereotactically injected, but the expression levels are lower like due to a lower virus titer in areas more distant to the injection site.

Note that the CRISPR-mediated *Clstn3* KO was instituted after synapse formation in the cerebellum is largely complete, and thus does not probe the initial establishment of synapses (*Hashimoto et al., 2009*), although cerebellar synapses are likely eliminated and reformed continuously throughout life. Analysis of the *Clstn3* CRISPR KO strategy for potential off-target effects demonstrated that at the sites most similar to the *Clstn3* target sequence, no mutations were detected (*Figure 2—figure supplement 1B-H*). Viewed together, these data indicate that the CRISPR-mediated *Clstn3* KO effectively ablates *Clstn3* expression in the cerebellar cortex.

## The cerebellar *Clstn3* KO causes major impairments in motor learning

To explore the functional consequences of the *Clstn3* KO in Purkinje cells, we analyzed its behavioral effects. Mice with the cerebellar *Clstn3* KO exhibited a normal open field behavior, but a striking impairment in motor learning as assayed with the rotarod task (*Figure 3A–G*). Moreover, the gait of cerebellar *Clstn3* KO mice was significantly altered with decreased stride lengths and increased overlap of fore- and hind-paw positions (*Figure 3H–K*). However, the cerebellar *Clstn3* KO mice displayed normal beam walk abilities and normal social behaviors, suggesting that the observed motor impairments are selective (*Figure 3L–R*). The motor coordination deficit in *Clstn3* CRISPR KO mice is consistent with phenotypes observed in constitutive *Clstn3* KO mice (https://www.mousephenotype.org) (*Dickinson et al., 2016*), suggesting that our CRISPR KO approach is reliable and specific. Given that *Clstn3* is predominantly expressed in Purkinje cells in the cerebellar cortex and that the AAVs used for the CRISPR KO primarily infect Purkinje cells but not other cells in the cerebellar cortex (*Figure 1*, *Figure 1—figure supplement 1*), the motor learning and gait deficits we observed after the *Clstn3* KO were likely due to the deletion of *Clstn3* in Purkinje cells. However, it should be noted that contributions of other cell types cannot be ruled out.

## The cerebellar *Clstn3* KO suppresses inhibitory synapse numbers in the cerebellar cortex

To test whether the *Clstn3* KO impaired motor behaviors by a synaptic mechanism as suggested by current views on calsyntenin functions (see Introduction), we first examined inhibitory synapses on Purkinje cells, which are the only postsynaptic targets for all excitatory and inhibitory inputs in the cerebellar cortex. We quantified the inhibitory synapse density in the cerebellar cortex using immunohistochemistry for vGAT and GABA-A$_{\alpha 1}$ receptors, which are pre and postsynaptic markers for inhibitory synapses in Purkinje cells (*Figure 4A–H*).

The CRISPR KO of *Clstn3* in the cerebellum robustly reduced the inhibitory synapse density in the molecular layer, Purkinje-cell layer, and granule-cell layer of the cerebellar cortex as measured both with the pre- and the postsynaptic marker. The most extensive decrease (~60–70%) was observed in the deep molecular layer (*Figure 4B and F*), where we also detected a significant reduction (~25%) in the size of vGAT-positive puncta, which correspond to the presynaptic vesicle clusters of inhibitory synapses (*Figure 4C*). vGAT- and GABA-A$_{\alpha 1}$ receptor-positive synapses in Purkinje cell and granule cell layers were less affected, but still exhibited a robust decrease (~20% reduction; *Figure 4D and H*).

The decline in inhibitory synapse density raises the question whether inhibitory synaptic transmission is suppressed. Therefore, we recorded miniature inhibitory postsynaptic currents (mIPSCs) from Purkinje cells in the presence of tetrodotoxin (*Figure 5*). The *Clstn3* KO produced a large decrease in mIPSC frequency (~60%), without changing the mIPSC amplitude (*Figure 5A–C*). Moreover, the *Clstn3*

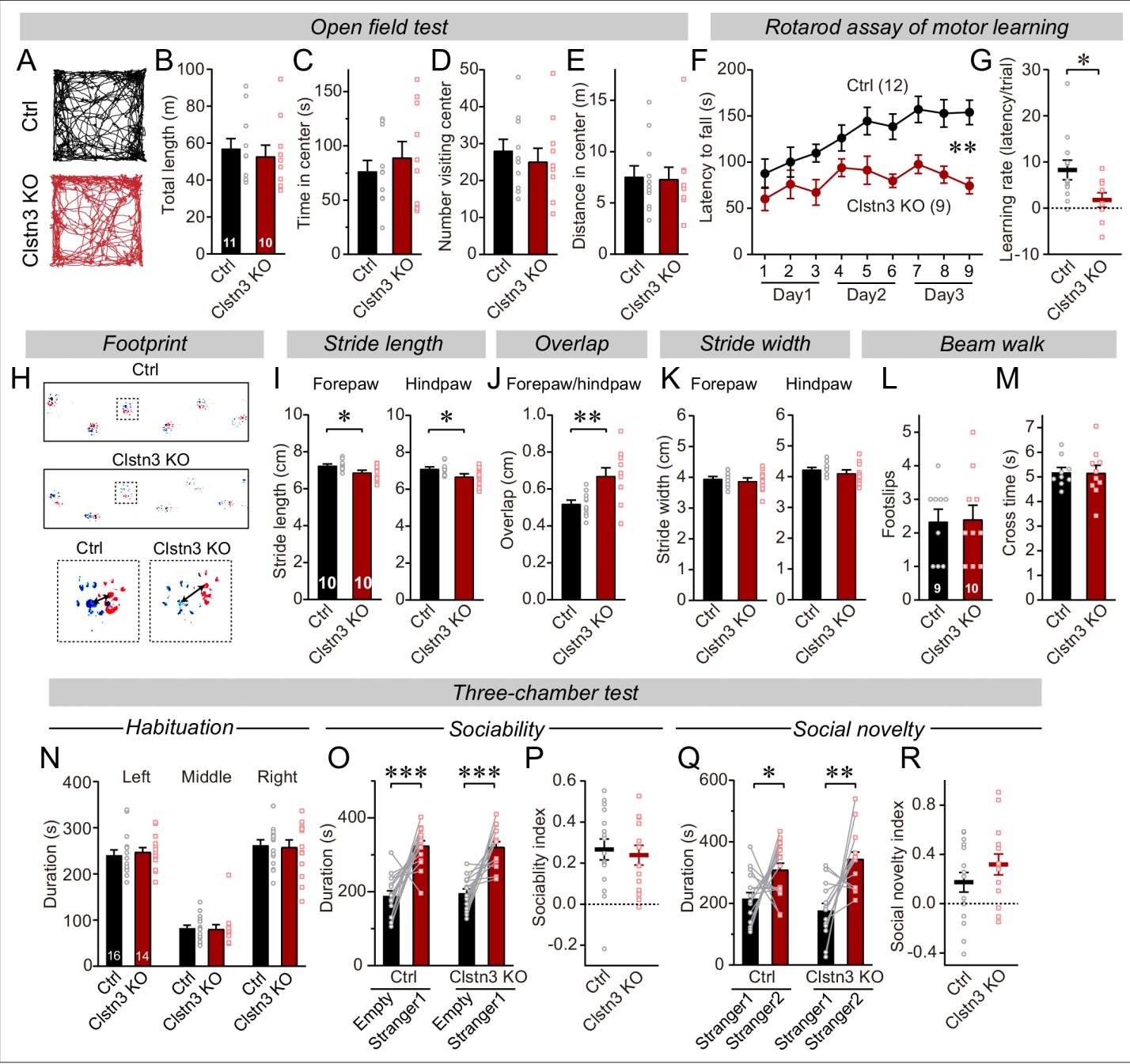

**Figure 3.** The CRISPR/Cas9-mediated deletion of *Clstn3* in cerebellum impairs motor learning and gait performance. (**A–E**) The CRISPR-mediated *Clsn3* KO had no effects on the open field behaviors. (**A**, representative tracks for both control and *Clstn3* KO; **B**, total length travelled (m); **C**, time spent in center; **D**, number of the mice visiting center; **E**, distance in center). (**F & G**) The CRISPR-mediated *Clstn3* KO in the cerebellum severely impairs motor learning as analyzed by the rotarod assay (**F**, rotarod learning curve; **G**, slope of rotarod curve used as an index of the learning rate). (**H–K**) *Clstn3* deletion significantly impaired gait performance. (**H**, upper images are representative footprints from both control and *Clstn3* KO group, the lower shows footprints of forepaw and hindpaw placements; I, stride length of forepaw and hindpaw from both control and *Clstn3* KO groups; J, distance between the center of forepaw and hindpaws; K, stride width of forepaw and hindpaw from both groups). (**L & M**) Beam walk test reveal no difference between the control group and the *Clstn3* KO group. (**L**, footslips; **M**, cross times).(**N–R**) CRISPR deletion of *Clstn3* in the cerebellum in vivo does not affect social behaviors. (**N**, control and *Clstn3* KO mice exhibited the same exploration behavior of the left and right chambers during the habituation period O and P, test mice were exposed to a non-familiar 'stranger' mouse in one of the outer chambers **O**, the time that the test mouse spent in the chambers with empty cup or 'stranger1' mouse; **P**, sociability index). **Q & R**, test mice were given the choice between exploring a 'stranger 1' mouse to which it was previously exposed, or a 'stranger 2' mouse that is novel, both control and cerebellar *Clstn3* KO mice prefer the novel mouse for interactions (**Q**, the same as **O**, except empty cup has 'stranger2'; **R**, social novelty index). All data are means ± SEM. Statistical analyses were performed

*Figure 3 continued on next page*

*Figure 3 continued*

using double-tailed and unpaired t-test for panels B-E, G, I-M, N, P, and R. Double-tailed paired t-tests were applied to analyze panel O and Q, *p < 0.05, **p < 0.01, ***p < 0.001. Repeat-measures ANOVA was applied for rotarod curve in panel F ($F_{(1, 19)}$ = 11.791, **p < 0.01). Numbers of animals for each experiment are indicated in graphs.

KO increased the rise but not decay times of mIPSCs (*Figure 5D*). Measurements of the Purkinje cell capacitance and input resistance showed that the *Clstn3* deletion did not produce major changes, demonstrating that it did not globally alter Purkinje cell properties (*Figure 5—figure supplement 1A*).

In the cerebellar cortex, basket and stellate cells are derived independently via distinct cell lineages, and gradually migrate to the proximal and distal layers of the cerebellar cortex, respectively

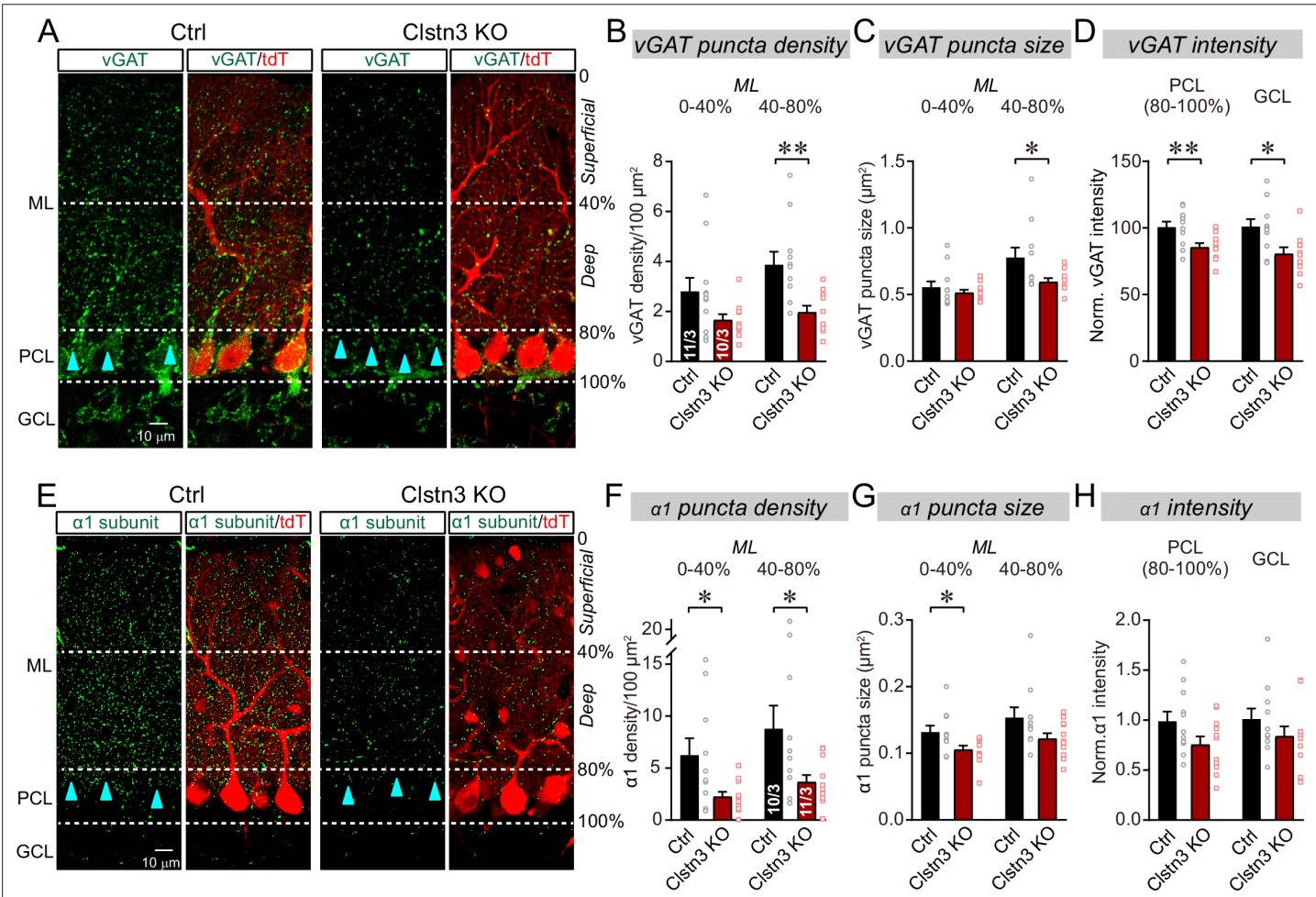

**Figure 4.** The *Clstn3* KO decreases inhibitory synapse numbers in the cerebellar cortex as revealed by immunocytochemistry for pre- and post-synaptic markers. (**A**) Representative confocal images of cerebellar cortex sections imaged for presynaptic vGAT and for tdTomato. Sections are from mice in which the cerebellar cortex was infected with control AAVs (Ctrl) or AAVs that induce the CRISPR-mediated *Clstn3* KO (green, vGAT; red, AAV-encoded tdTomato signal; ML, molecular layer; PCL, Purkinje cell layer; GCL, granule cell layer). Calibration bar applies to all images. (**B–D**) Quantifications of vGAT-positive synaptic puncta demonstrating that the *Clstn3* KO suppresses the number of inhibitory synapses in the cerebellar cortex (B and C, density and size of vGAT-positive puncta in the molecular layer (ML) of the cerebellar cortex (separated into deep (40–80%) and superficial areas (0–14%)); D, vGAT-staining intensity in the Purkinje cell layer (PCL) and granule cell layer (GCL) of the cerebellar cortex as a proxy of synapse density since individual vGAT-positive puncta cannot be resolved in these layers). (**E**) Representative confocal images of cerebellar cortex sections imaged for postsynaptic GABA-A$_{\alpha1}$ receptor subunits and for tdTomato (green, GABA-A$_{\alpha1}$ receptor subunit; red, AAV-encoded tdTomato signal; labeling is the same as in A). (**F–H**) Quantifications of GABA-A$_{\alpha1}$ receptor-positive synaptic puncta independently confirmed that the *Clstn3* KO suppresses the number of inhibitory synapses in the cerebellar cortex (F-H are the same as B-D, but for GABA-A$_{\alpha1}$ receptor staining instead of vGAT staining). Data are means ± SEM (numbers of sections/mice analyzed are indicated in bar graphs). Statistical analyses were performed using two tailed unpaired t-tests, with *p < 0.05, **p < 0.01.

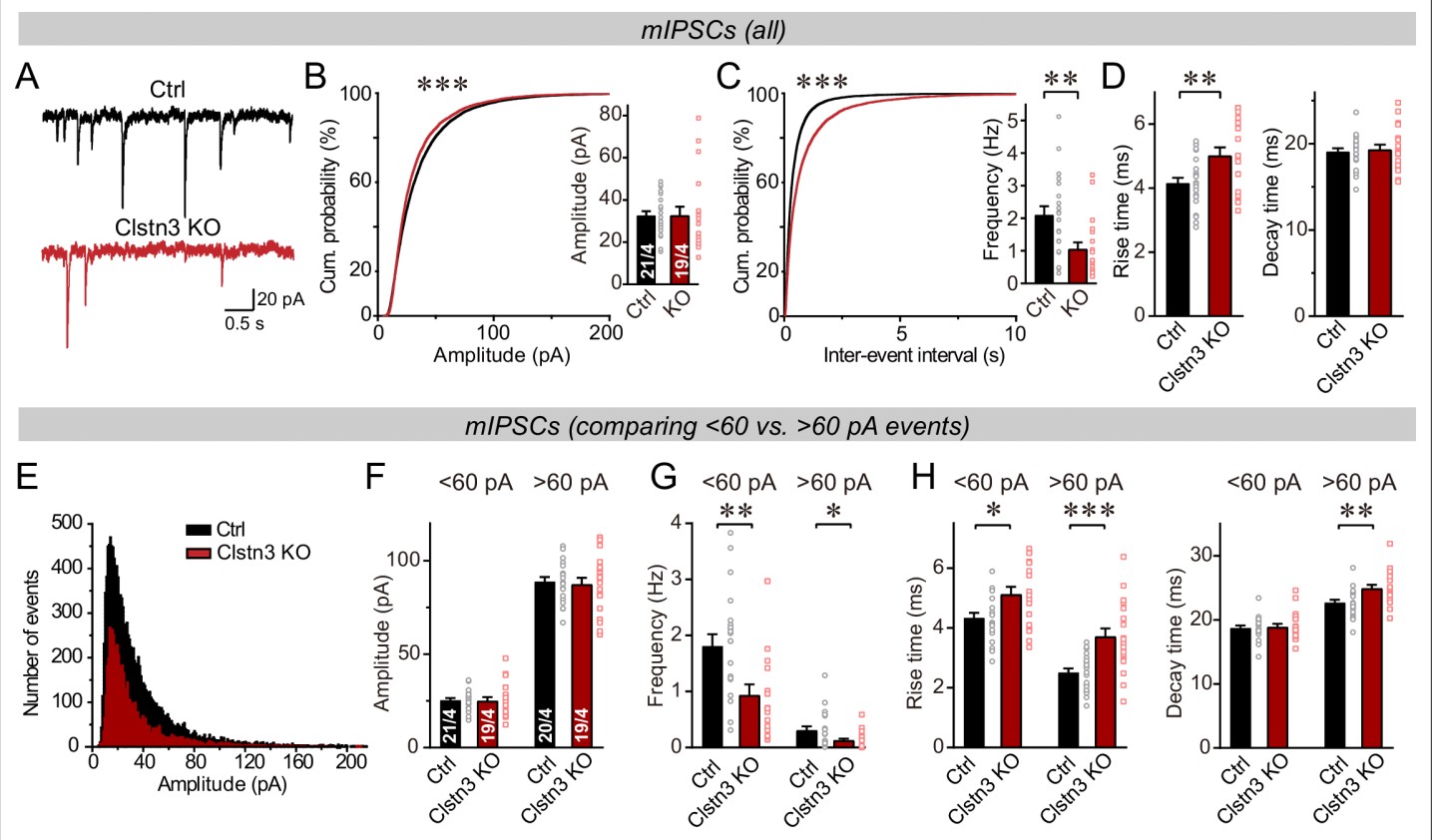

**Figure 5.** The *Clstn3* KO decreases spontaneous inhibitory synaptic 'mini' events in Purkinje cells. (**A–C**) The *Clstn3* KO decreases the frequency but not the amplitude of mIPSCs (A, representative traces; B, left, cumulative probability plot of the mIPSC amplitude right, average amplitude; C, left, cumulative probability plot of the mIPSC inter-event interval right, average frequency). (**D**) The *Clstn3* KO increases the rise but not decay times of mIPSCs. (**E**) Plot of the number of mIPSC events vs. amplitude exhibiting a normal distribution. (**F–H**) The *Clstn3* KO similarly impairs mIPSCs with a larger ( > 60 pA) and a smaller amplitude ( < 60 pA), which in Purkinje cells are likely generated primarily by basket cell and stellate cell synapses, respectively (**F & G**, summary graphs for the mIPSC amplitude [**F**] and frequency [**G**] separately analyzed for high- and low-amplitude events; H, mIPSC rise [left] and decay times [right], separately analyzed for high- and low-amplitude events). All summary data are means ± SEM. Numbers of cells/mice analyzed are indicated in bar graphs. Statistical analyses were performed using unpaired t-tests (bar graphs with two groups) or Kolmogorov-Smirnov test (cumulative analysis), with *p < 0.05, **p < 0.01, ***p < 0.001.

The online version of this article includes the following figure supplement(s) for figure 5:

**Figure supplement 1.** The capacitance and membrane resistance of Purkinje cells are unaffected by the *Clstn3* KO.

**Figure supplement 2.** Analysis of the kinetics of large ( > 60 pA) and smaller ( < 60 pA) mIPSCs confirm that larger mIPSCs, which are presumably generated by basket-cell synapses closer to the soma, have faster rise but slower decay times.

(*Zhang and Goldman, 1996*; *Wang et al., 2005*). Basket cells form inhibitory synapses on the axon initial segment, soma and proximal dendrites of Purkinje cells, whereas stellate cells form inhibitory synapses on the more distant dendritic domains. Because these cells inhibit Purkinje cells at different subcellular locations, their synaptic outputs differentially shape the Purkinje cell activity (*Brown et al., 2019*; *Nakayama et al., 2012*). As an approximate whether either basket or stellate cell synapses might be preferentially impaired by the *Clstn3* deletion, we analyzed mIPSCs as function of their amplitudes, with the notion that more distance inhibitory synapses derived from stellate cells would have lower amplitudes. First, we plotted the mIPSC amplitudes in a normal distribution (*Figure 5E*), which revealed that the majority of mIPSCs ( > 90%) exhibit amplitudes of <60 pA. Next, we separately analyzed mIPSCs with amplitudes of >60 pA and <60 pA, of which the >60 pA mIPSCs likely mostly represent basket cell mIPSCs, whereas the <60 pA mIPSCs are composed of a mixture of stellate and baseket cell mIPSCs (*Nakayama et al., 2012*). Consistent with a predominant localization of the larger mIPSCs to proximal basket cell synapses and of the smaller mIPSCs to more distant stellate and basket cell synapses, the former displayed a faster

rise and slower decay kinetics than the latter (*Figure 5—figure supplement 2*). Importantly, both classes of mIPSCs exhibited similar impairments in frequency, although the changes were slightly more pronounced for larger mIPSCs (*Figure 5F–H*). Consistent with the morphological data, these data suggest a decrease in both basket and stellate cell synapses, with the former probably more severely impacted than the latter by the *Clstn3* KO (*Figure 4A–D*). Note, however, that these analyses are only approximations, and that in the absence of paired recordings of large numbers of synapses, it is not possible to define the relative impairment of the two types of cerebellar inhibitory synapses accurately.

Does the decrease in inhibitory synapse density and mIPSCs cause a change in overall inhibitory synaptic strength? We examined evoked inhibitory synaptic responses, using extracellular stimulations of basket cell axons close to the Purkinje cell layer (*Figure 6A*). These measurements revealed a significant decrease (~40%) in IPSC amplitudes. The decrease in IPSC amplitude and mIPSC frequency is consistent with a loss of inhibitory synapses as suggested by immunohistochemistry (*Figure 4*), but could also be due to a decrease in release probability. However, we detected no major changes in the coefficient of variation, paired-pulse ratio, or kinetics of evoked IPSCs, suggesting that the release probability is normal (*Figure 6B–F*, *Figure 5—figure supplement 1B*). These data confirm the morphological results, together indicating that the *Clstn3* KO decreases inhibitory synapse numbers on Purkinje cells.

## The cerebellar *Clstn3* deletion increases excitatory parallel-fiber but not climbing-fiber synapse numbers

The decrease in inhibitory synapse numbers by the *Clstn3* KO is consistent with previous studies suggesting that *Clstn3* promotes synapse formation in the hippocampus, but the primary impairment identified in these studies was a decrease in excitatory, and not inhibitory, synapses (*Kim et al., 2020*; *Pettem et al., 2013*; *Ranneva et al., 2020*). We thus tested whether the *Clstn3* KO similarly affects excitatory synapse numbers in the cerebellum. Purkinje cells receive two different excitatory synaptic inputs with distinct properties: Parallel-fiber synapses that are formed by granule cells on distant Purkinje cell dendrites, and climbing-fiber synapses that are formed by inferior olive neurons on proximal Purkinje cell dendrites. Parallel-fiber synapses use the vesicular glutamate transporter vGluT1, whereas climbing-fiber synapses use the vesicular glutamate transporter vGluT2 (*Hioki et al., 2003*). Moreover, parallel-fiber synapses are surrounded by astrocytic processes formed by Bergmann glia, creating a tripartite synapse in which the glial processes contain high levels of GluA1 (*Baude et al., 1994*). Thus, to measure the effect of the *Clstn3* KO on excitatory Purkinje cell synapses, we analyzed cerebellar sections from control and cerebellar *Clstn3* KO mice by immunohistochemistry for vGluT1, vGluT2, GluA2, and GluA1 (*Figure 7*).

Confocal microscopy of cerebellar cortex sections immunolabeled for vGluT1 revealed that, surprisingly, vGluT1 staining was enhanced by the *Clstn3* deletion instead of being suppressed (*Figure 7A*). Because parallel-fiber synapses in the cerebellar cortex are so numerous that confocal microscopy cannot resolve individual vGluT1-positive synaptic puncta, we measured the overall vGluT1 staining intensity as a proxy for synapse density (*Zhang et al., 2015*). The cerebellar *Clstn3* deletion caused a robust increase (> 25%) in the vGluT1 staining intensity of both the superficial and the deep molecular layers of the cerebellar cortex (*Figure 7B*).

The potential increase in parallel-fiber synapses induced by the *Clstn3* KO, suggested by the enhanced vGluT1 staining intensity, was unexpected. This prompted us to examine the staining intensity for GluA2, the predominant postsynaptic AMPA-receptor of parallel-fiber synapses, which directly reflects the density of parallel-fiber synapses on Purkinje cells. The cerebellar *Clstn3* KO induced an even more striking increase (~80%) in GluA2 staining in Purkinje cells than that observed for vGluT1 staining (*Figure 7C and D*), demonstrating that immunohistochemistry for both pre- and postsynaptic parallel-fiber synapses reveal an increase in parallel-fiber synapses induced by the cerebellar *Clstn3* KO.

To further validate the observed increase in parallel-fiber synapse density, we next measured the levels of GluA1 as an astroglial marker of tripartite parallel-fiber synapses (*Figure 7E*; *Baude et al., 1994*). Again, the cerebellar *Clstn3* KO induced a significant enhancement (~25%) in synaptic GluA1 staining intensity (*Figure 7F*), consistent with the increase in vGluT1 and GluA2 staining intensity. Thus, immunohistochemistry for three different synaptic markers present on distinct synaptic compartments

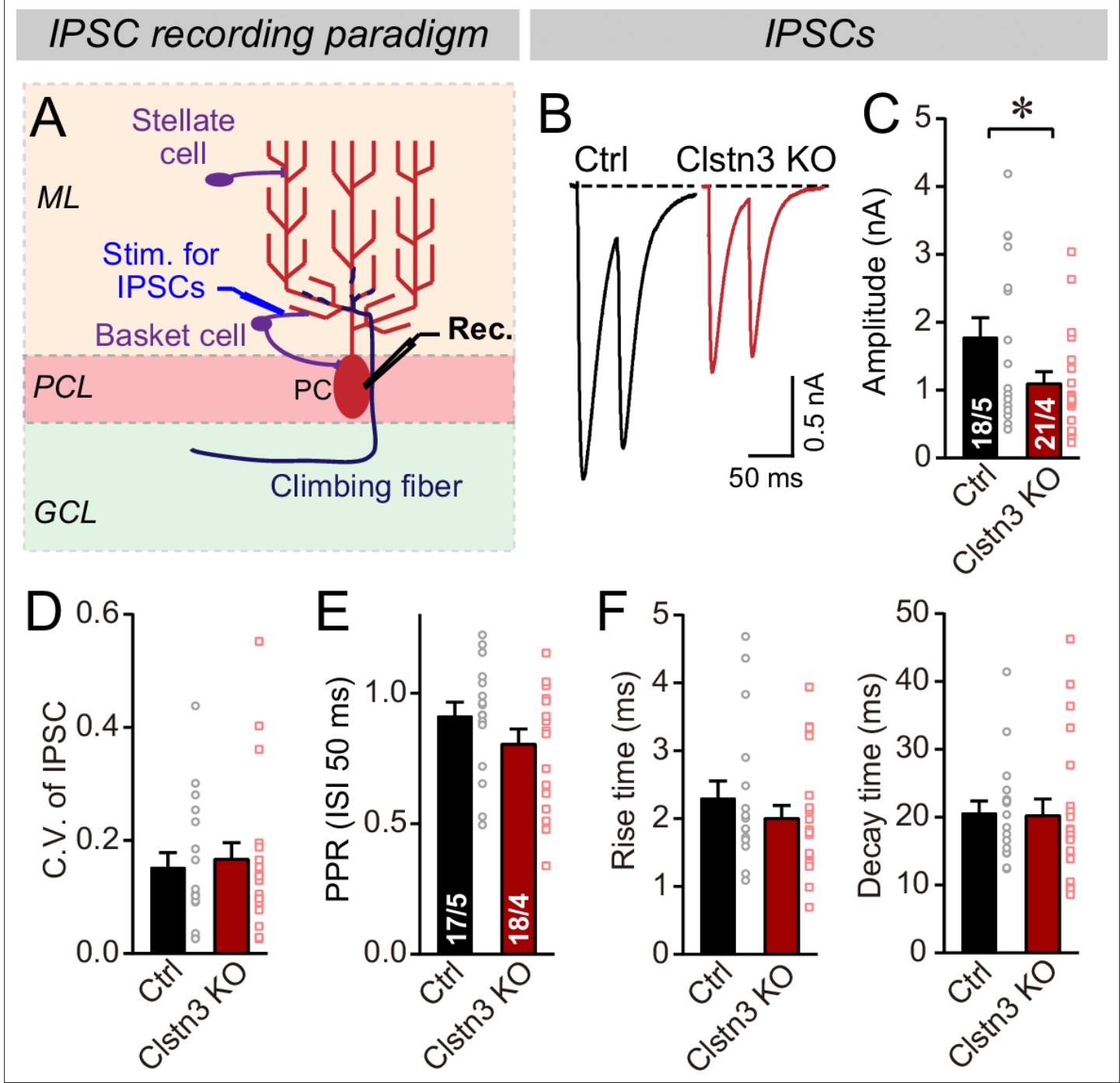

**Figure 6.** The *Clstn3* KO decreases evoked inhibitory synaptic responses in Purkinje cells. (**A**) Experimental design for recordings of IPSCs evoked by stimulation of basket cell axons (ML, molecular layer; PCL, Purkinje cell layer; GCL, granule cell layer; PC, Purkinje cell; Rec., recording patch pipette). (**B & C**) The *Clstn3* KO decreases the amplitude of evoked basket-cell IPSCs (B, representative traces of pairs of evoked IPSCs with a 50ms inter-stimulus interval; (**C**) summary graphs of the amplitude of the first IPSC). (**D & E**) The *Clstn3* KO in Purkinje cells does not affect the release probability at inhibitory synapses as judged by the coefficient of variation (**D**) and the paired-pulse ratio with an interstimulus interval of 50ms (**E**) of evoked IPSCs. (**F**) The *Clstn3* KO in Purkinje cells has no significant effect on IPSC kinetics (left, rise times; right, decay times of evoked ISPCs). All summary data are means ± SEM. Numbers of cells/mice analyzed are indicated in bar graphs. Statistical analyses were performed using unpaired t-tests, with *$p < 0.05$.

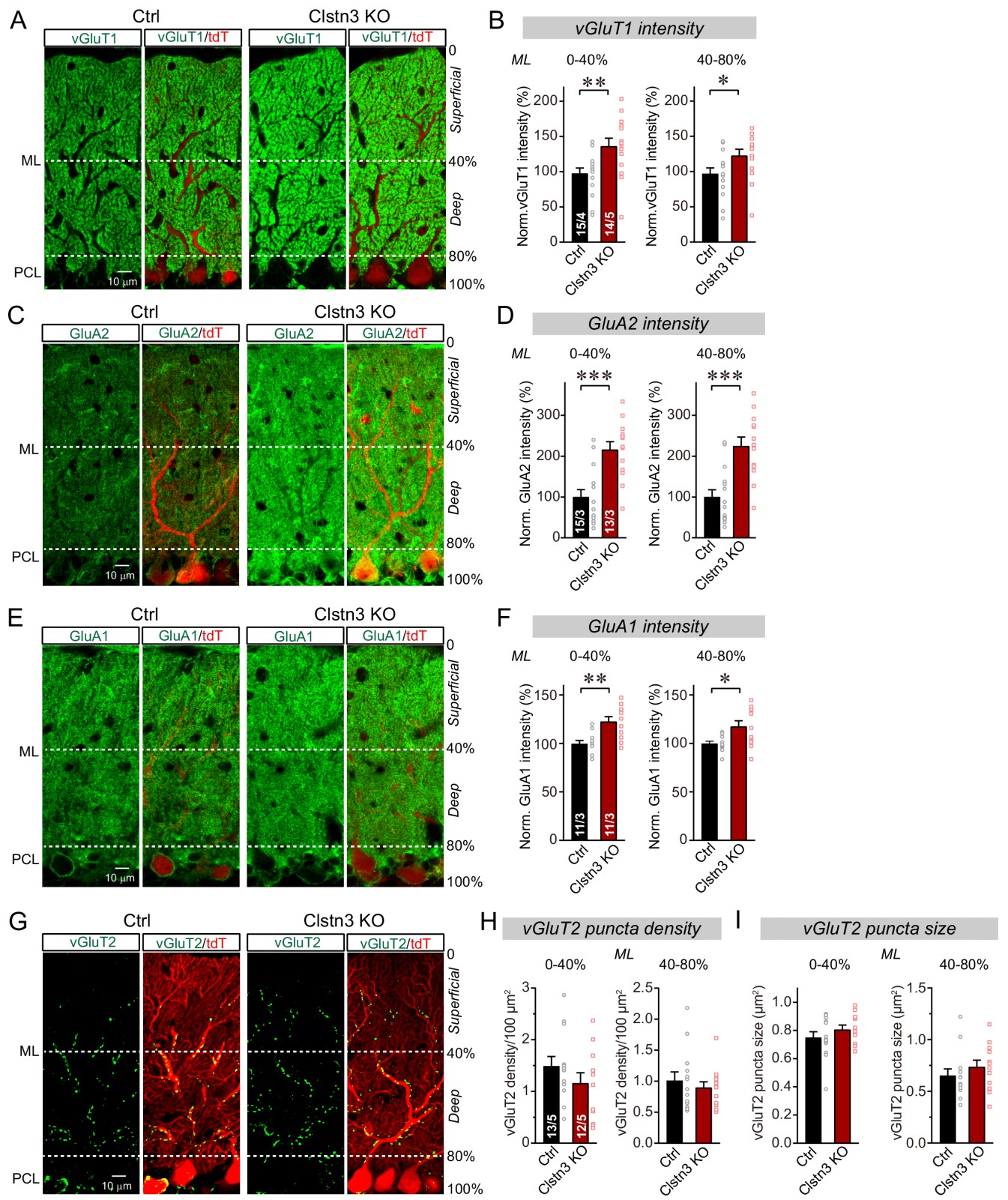

**Figure 7.** CRISPR-mediated *Clstn3* deletion in the cerebellar cortex increases parallel-fiber excitatory synapse numbers without changing climbing-fiber synapse numbers. (**A & B**) Immunostaining of cerebellar cortex sections an with antibody to vGluT1 as a presynaptic marker for parallel-fiber synapses reveals a significant increase (**A**, representative confocal images from control and *Clstn3* KO mice [green vGluT1; red, tdTomato]; (**B**) summary graphs of the vGluT1 staining intensity in the superficial (0–40%) and deep (40–80%) molecular layers of the cerebellar cortex). Note that the staining intensity

*Figure 7 continued on next page*

*Figure 7 continued*

is used as a proxy for synapse density since individual parallel-fiber synapse puncta cannot be resolved. (**C & D**) Immunostaining with an antibody to GluA2 as a postsynaptic marker for parallel-fiber synapses confirms the robust increase in parallel-fiber synapse abundance observed with vGluT1 staining (**E**, representative confocal images from control and *Clstn3* KO mice [green, GluA2; red, tdTomato]; F, summary graphs of the GluA2 staining intensity in the superficial (0–40%) and deep (40–80%) molecular layers of the cerebellar cortex). (**E & F**) Immunostaining with antibody to GluA1, an astroglial marker for tripartite parallel-fiber synapses containing Bergmann glia processes, also uncovers a significant increase in staining intensity (**C**, representative confocal images from control and *Clstn3* KO samples [green vGluT1; red, tdTomato]; (**D**) summary graphs of the GluA1 staining intensity in the superficial (0–40%) and deep (40–80%) molecular layers of the cerebellar cortex). (**G–I**) Immunostaining for vGluT2 as a marker for climbing-fiber synapses in cerebellar cortex fails to detect a *Clstn3* KO-induced change (**G**, representative confocal images [green, vGluT2; red, tdTomato]; (**H & I**) summary graphs of the density (**H**) and size (**I**) of vGluT2-positive synaptic puncta in the superficial (0–40%) and deep (40–80%) molecular layers of the cerebellar cortex). All numerical data are means ± SEM; numbers of sections/mice analyzed are indicated in the first bar graphs for each experiment. Statistical significance was assessed by unpaired Student's t-test (*$p < 0.05$, **$p < 0.01$, ***$p < 0.001$).

supports the conclusion that the cerebellar *Clstn3* KO causes an increase in the number of parallel-fiber synapses.

Finally, we analyzed the density of climbing-fiber synapses by staining cerebellar sections for vGluT2, but detected no significant effect of the *Clstn3* KO. Different from parallel-fiber synapses that contain vGluT1, climbing-fiber synapses labeled with antibodies to vGluT2 are readily resolved by confocal microscopy (*Figure 7G*). The number and size of synaptic puncta identified with vGluT2 antibodies were not altered by the *Clstn3* KO, although there was a slight trend towards a decrease in climbing-fiber synapse density (*Figure 7H, I*). These observations suggest that the enhancement of parallel-fiber synapse density by the *Clstn3* KO is specific for this type of synapse.

## The *Clstn3* KO increases the spine density of Purkinje cells without changing their dendritic arborization

It is surprising that the cerebellar *Clstn3* KO appears to increase the parallel-fiber synapse density, as one would expect a synaptic adhesion molecule to promote but not to suppress formation of a particular synapse. The parallel-fiber synapse increase is likely not a homeostatic response to the loss of inhibitory synapses because such a response, which would aim to maintain a constant excitatory/inhibitory balance, should produce a decrease, but not an increase, in parallel-fiber synapses in response to a decrease in inhibitory inputs (*Jörntell, 2017*; *Li et al., 2019*; *Chowdhury and Hell, 2018*). The increase in parallel-fiber synapse numbers is also unexpected given previous results showing that in hippocampal CA1 neurons, the *Clstn3* KO decreases excitatory synapse numbers (*Kim et al., 2020*; *Pettem et al., 2013*). To independently examine the density of parallel-fiber synapses and to also test whether the *Clstn3* KO has a notable effect on the dendritic arborization of Purkinje cells, we analyzed Purkinje cells morphologically. The dendritic spine density of Purkinje cells is a useful, but imperfect, proxy of parallel-fiber synapses because in a normal brain, all spines contain parallel-fiber synapses and all parallel-fiber synapses are on spines (*Sotelo, 1975*). However, some mutations, such as the *Cbln1* deletion (*Yuzaki and Ariescu, 2017*), cause a partial loss of parallel-fiber synapses and induce the appearance of 'naked' spines. As a result, the spine density of Purkinje cells by itself is not a perfect measure of parallel-fiber synapses. It needs to be complemented by other approaches, such as the immunocytochemical quantification of synapses that was described above, although no condition has been shown to cause an increased spine density due to an increase of naked spines (*Südhof, 2017*).

We filled individual Purkinje cells in acute slices with biocytin via a patch-pipette, and fully reconstructed six Purkinje cells from control and *Clstn3* KO mice to analyze their dendritic structure and their spine density (*Figure 8A*, *Figure 8—figure supplement 1A and B*). These reconstructions revealed a trend toward an increased dendrite length in *Clstn3*-deficient Purkinje cells without a significant change in dendritic architecture, demonstrating that the *Clstn3* KO does not impair the overall structure of Purkinje cells (*Figure 8B*). Quantification of dendritic spines uncovered in *Clstn3*-deficient Purkinje cells a robust increase (~30%) in the density of spines in the superficial area of the cerebellar cortex, and a trend toward an increase in the deep area of the cerebellar cortex (*Figure 8C–F*). The increase in spine density was particularly pronounced for thin spines (*Figure 8G*; *Figure 8—figure supplement 1C and D*). These findings provide independent evidence that the *Clstn3* KO increases the parallel-fiber synapse density, and precisely mirror those obtained by analyzing the vGluT1-,

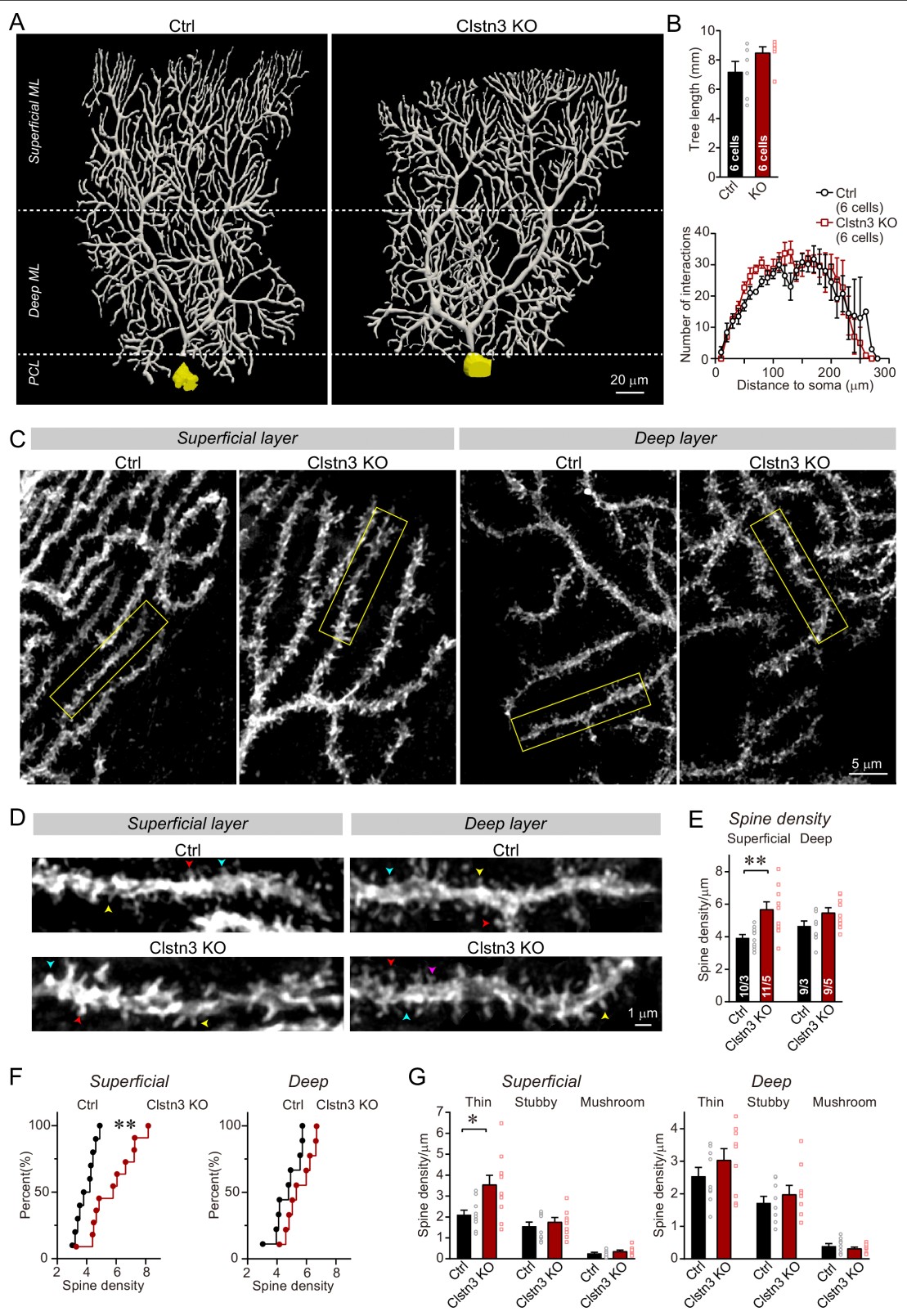

**Figure 8.** Morphological analysis of individual Purkinje cells reveals that the *Clstn3* KO robustly increases the dendritic spine density of Purkinje cells without significantly altering their dendritic arborization. (**A & B**) Biocytin filling of individual Purkinje cells via a patch pipette demonstrates that the *Clstn3* KO does not significantly change the overall dendritic architecture of Purkinje cells (**A**, representative images of Purkinje cell dendritic trees for control and *Clstn3* KO mice after 3D reconstruction [for more images, see , *Figure 8—figure supplement 1*]; **B**, quantifications of the dendritic tree

*Figure 8 continued*

length [top] or dendritic arborization using Scholl analysis [bottom]). (**C–F**) The *Clstn3* KO increases the density of dendritic spines of Purkinje cells in the superficial part of the cerebellar cortex (**C & D**, representative images of spiny dendrites at low and high magnifications, respectively; [blue, red, and yellows arrowheads mark different spine types]; (**E & F**) summary graph [E] and cumulative distribution of the spine density [F]). (**G**) The *Clstn3* KO in Purkinje cells increases preferentially the density of thin spines in the superficial part of the cerebellar cortex, based on a morphological classification of spine types into thin, stubby and mushroom spines. All data in B, E, and G are means ± SEM; 6 control and *Clstn3* KO Purkinje cells were reconstructed for B; numbers in the first bars of E indicate the number of cell/animal analyzed for E-G. Statistical significance (*p < 0.05; **p < 0.01) in B and G was assessed by an unpaired t-test, and in E by one-way ANOVA ($F_{(3, 35)}$ = 5.693, p = 0.003), followed by Tukey's post hoc comparisons for control and *Clstn3* KO groups.

The online version of this article includes the following figure supplement(s) for figure 8:

**Figure supplement 1.** Images of individual reconstructed biocytin-filled Purkinje cells, and further quantifications of the morphological properties of spines from control and *Clstn3* KO Purkinje cells.

GluA2-, and GluA1-staining intensity of the cerebellar cortex (*Figure 6A–D*). Note that in some mouse mutants of synaptic proteins – for example those deleting cerebellin-1 or GluD2 - 'naked' spines are observed because in these mutants, synapses are initially formed but presynaptic terminals are partly degraded (*Südhof, 2017*). As mentioned above, the increase in spine density we observed is thus not by itself a reliable indication of an increase in synapse density, since it is conceivable that the added spines are 'naked'. However, since the increase in spine density after the *Clstn3* deletion correlated well with the similar increases in both presynaptic and postsynaptic markers of parallel-fiber synapses and in Bergmann glia markers associated with tripartite parallel-fiber synapses, the most plausible explanation for the increase in spine density is an increase in synapse density. To further support this hypothesis, we performed the following electrophysiological experiments that directly measure the synaptic transmissions.

## The *Clstn3* KO increases parallel-fiber but not climbing-fiber synaptic transmission

The increase in parallel-fiber synapses could be due to a true enhancement of parallel-fiber synapse formation, or a homeostatic reaction to a large decrease in parallel-fiber synapse function. To clarify this question, we analyzed parallel-fiber synapse function by electrophysiology, and compared it to climbing-fiber synapse function as an internal control since climbing-fiber synapse numbers are not changed by the cerebellar *Clstn3* KO.

We first monitored spontaneous miniature synaptic events (mEPSCs) in the presence of tetrodo-toxin. We observed an increase in mEPSC amplitudes (~25%) and frequency (~15%) in *Clstn3* KO neurons, without a notable change in mEPSC kinetics (*Figure 9A-D*, *Figure 5—figure supplement 1C*). The majority of mEPSCs in Purkinje cells are derived from parallel-fiber synapses. Because of the large dendritic tree of Purkinje cells, synapses on distant dendrites produce slower and smaller mEPSCs than synapses on proximal dendrites (*Zhang et al., 2015*). To preferentially analyze mEPSCs derived from parallel-fiber synapses (whose density is increased morphologically), we used a approach similar to that we employed for mIPSCs and examined slow mEPSCs with rise times of >1ms separately. These slow mEPSCs are mostly generated by parallel-fiber synapses on distant dendrites, although a smaller contribution of mEPSCs derived from climbing-fiber synapses cannot be excluded (*Nakayama et al., 2012*; *Yamasaki et al., 2006*). The results of the analysis of slower mEPSCs were the same as for total mEPSCs, confirming that the *Clstn3* KO increases parallel-fiber synaptic activity (*Figure 9E–H*).

Next, we measured evoked parallel-fiber EPSCs, using input-output curves to correct for varia-tions in the placement of the stimulating electrode (*Figure 10A*, *Figure 5—figure supplement 1E*). Consistent with the morphological and mEPSC data, the *Clstn3* KO robustly enhanced parallel-fiber synaptic responses (~60% increase) (*Figure 10B–D*). This finding suggests that the *Clstn3* KO not only increases the density of parallel-fiber synapses, but also renders these synapses more efficacious. The increased strength of parallel-fiber synaptic transmission was not due to a change in release proba-bility because neither the coefficient of variation nor the paired-pulse ratios of parallel-fiber EPSCs were altered (*Figure 10E–G*). Thus, the increase in parallel-fiber EPSC amplitudes is consistent with enhancement of the vGluT1, GluA1, and GluA2 staining intensity, providing further evidence that the *Clstn3 KO* elevates parallel-fiber synapse numbers.

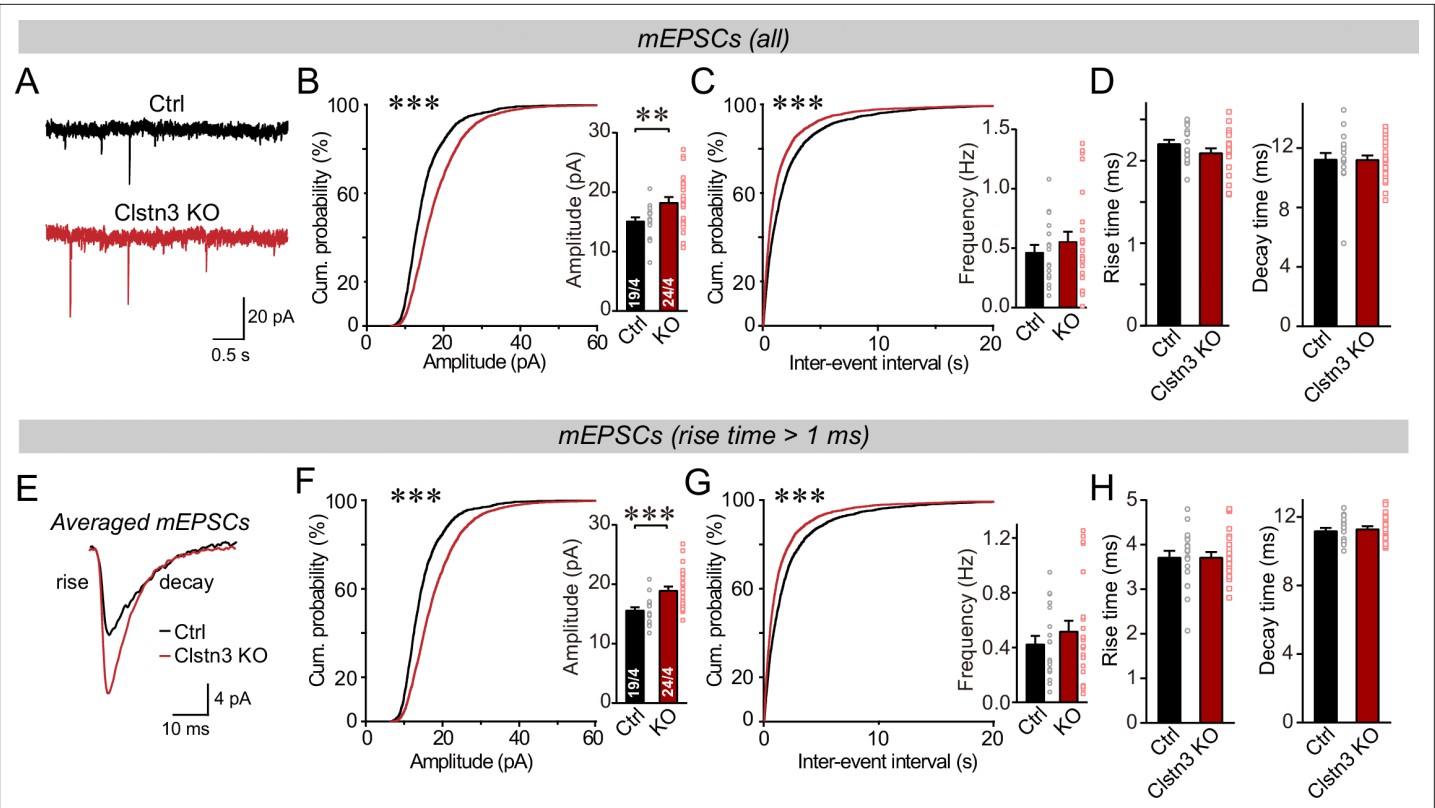

**Figure 9.** The *Clstn3* KO in cerebellar cortex increases the amplitude and frequency of parallel-fiber mEPSCs. (**A–C**) The cerebellar *Clstn3* KO increases the amplitude and frequency of mEPSCs in Purkinje cells (A, representative traces; B, left, cumulative probability plot of the mEPSC amplitude right, average amplitude; C, cumulative probability plot of the mEPSC inter-event interval [inset, average frequency]). (**D**) The cerebellar *Clstn3* KO has no effect on mEPSC kinetics (left, mEPSC rise times; right, mEPSC decay times). (**E**) Expanded traces of averaged mEPSCs to illustrate the kinetic similarity of control and *Clstn3* KO events with a change in amplitude. (**F & G**) mEPSCs with slow rise times ( > 1ms) and that are likely primarily derived from parallel-fiber synapses exhibit the same phenotype as the total mEPSCs (same as B and C, but for mEPSCs with slow rise times). (**H**) The cerebellar *Clstn3* KO has no effect on the kinetics of mEPSCs with slow rise times (left, mEPSC rise times; right, mEPSC decay times). All numerical data are means ± SEM. Statistical significance with two groups was assessed by unpaired t-test (*p < 0.05, **p < 0.01), with the number of cells/mice analyzed indicated in the first bar graphs for each experiment. Cumulative analysis was done with Kolmogorov-Smirnov test (***p < 0.001).

In contrast to parallel-fiber EPSCs, climbing-fiber EPSCs exhibited no *Clstn3* KO-induced alteration. Specifically, the amplitude, paired-pulse ratio, and kinetics of climbing-fiber EPSCs in control and *Clstn3* KO Purkinje cells were indistinguishable (*Figure 10H-L*, *Figure 5—figure supplement 1D*). These findings are consistent with the lack of a change in vGluT2-positive synaptic puncta analyzed morphologically (*Figure 7G–I*). Viewed together, these data suggest that *Clstn3 KO* produces an increase in excitatory parallel-fiber, but not climbing-fiber, synapses.

## The cerebellar *Clstn3* KO phenotype is specifically due to the deletion of *Clstn3* from Purkinje cells

The most plausible interpretation of our results up to this point is that suppression of *Clstn3* expression in Purkinje cells causes a surprising pair of opposite phenotypes: A decrease in inhibitory synapses and an increase of excitatory parallel-fiber synapses. That these phenotypes are caused by the deletion of *Clstn3* specifically in Purkinje cells is indicated by the exclusively high expression of *Clstn3* in these neurons, the low expression of Clstn3 in other cerebellar cell types, and the preferential infection of Purkinje cells by the AAV serotype we were using (*Figure 1*, *Figure 1—figure supplement 1*, *Figure 2—figure supplement 2*). However, the CRISPR-expressing AAVs we used to delete *Clstn3* likely partially infect other cerebellar cell types. Since granule cells, basket cells, and stellate cells also express *Clstn3*, albeit at low levels, it is possible that the two opposite synaptic phenotypes we observe could be due to the deletion of *Clstn3* in other cerebellar neurons in addition to Purkinje cells.

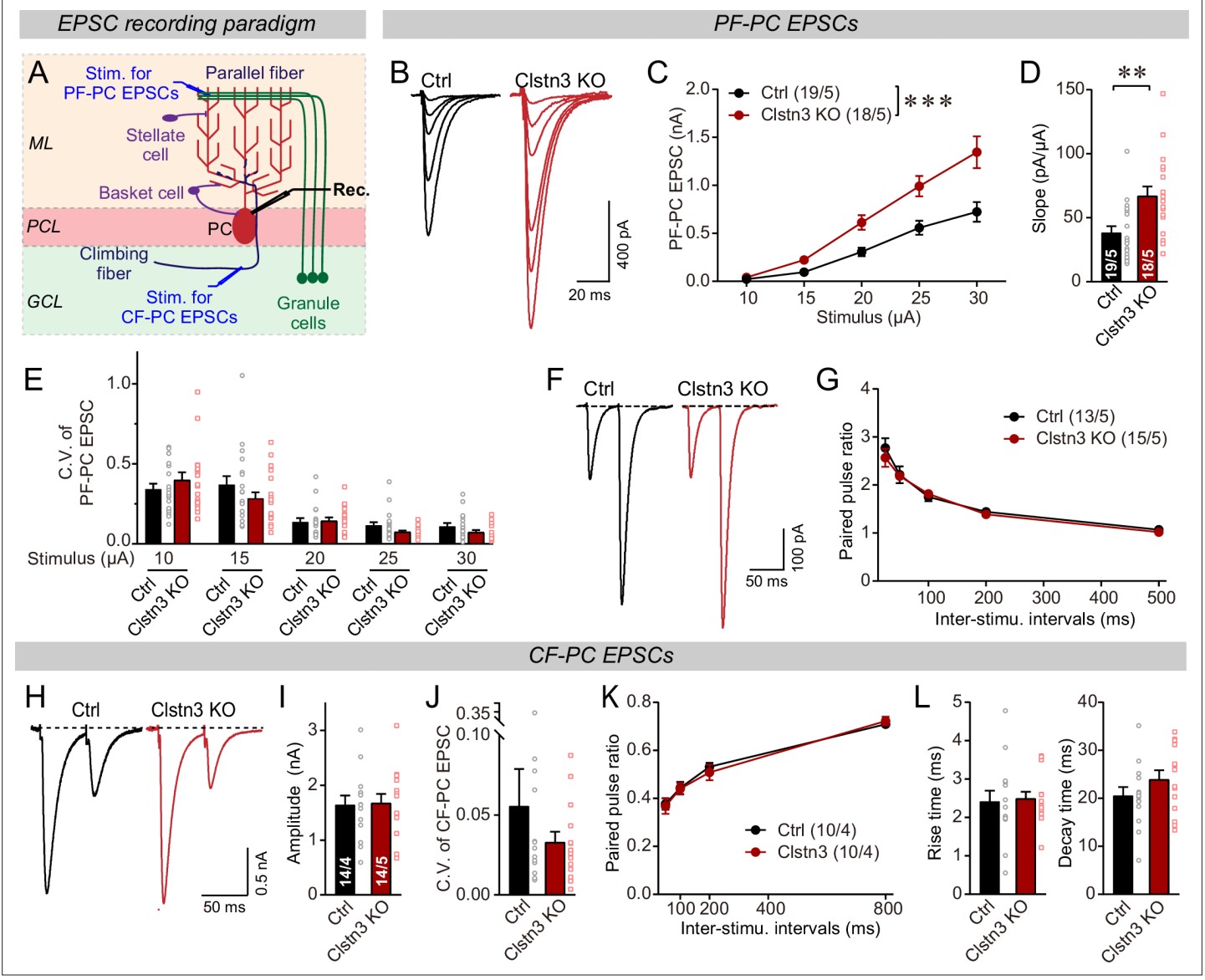

**Figure 10.** The *Clstn3* KO elevates the strength of parallel-fiber synapses without altering their release probability, but leaves climbing-fiber synapses unchanged. (**A**) Schematic of the recording configuration for monitoring evoked EPSCs induced by parallel-fiber (PF-PC) and climbing-fiber stimulation (CF-PC) in Purkinje cells. (**B–D**) The postsynaptic *Clstn3* KO robustly increases the input/output relation of parallel-fiber synapses (**B**, representative traces; **C**, input/output curve; **D**, summary graph of the slope of the input/output curve determined in individual cells). (**E–G**) The postsynaptic *Clstn3* KO in Purkinje cells has no effect on presynaptic release probability as assessed by monitoring the coefficient of variation of evoked EPSCs (**E**, separately analyzed for different stimulus strengths) or the paired-pulse ratio (**F**, sample traces; **G**, plot of the paired-pulse ratio of parallel-fiber EPSCs as a function of interstimulus interval). (**H–L**) The *Clstn3* KO has no effect on the amplitude, coefficient of variation, paired-pulse ratio, or kinetics of climbing-fiber synapse EPSCs, suggesting that it does not alter their properties (**H**, representative traces of climbing-fiber EPSCs elicited with an interstimulus interval of 50ms; **I & J**) amplitude (**I**) and coefficient of variation (**J**) of evoked climbing-fiber EPSCs; K, plot of the paired-pulse ratio of climbing-fiber EPSCs as a function of interstimulus interval; L, rise [left] and decay times [right] of evoked climbing-fiber EPSCs). All numerical data are means ± SEM. Statistical analyses were performed by two-way ANOVA followed by Tukey's post hoc correction (C, G, K; for C, $F_{(1, 150)}$ = 15.24, p < 0.0001) or unpaired t-test for experiments with two groups (**D, E, I, J, L**), with *p < 0.05, **p < 0.01.

In other words, our results do not rule out the possibility that part of the *Clstn3* KO phenotype is due to a partial deletion of *Clstn3* from other cerebellar cell types.

To address this important possibility, we performed a sparse CRISPR-mediated deletion of *Clstn3* in Purkinje cells. We infected the cerebellar cortex of Cas9 mice at P21 with low titers of lentiviruses expressing the same sgRNAs and tdTomato as the AAVs used for the earlier experiments, and analyzed the neurons 4–5 weeks later (*Figure 11A*). The same control infections were employed as

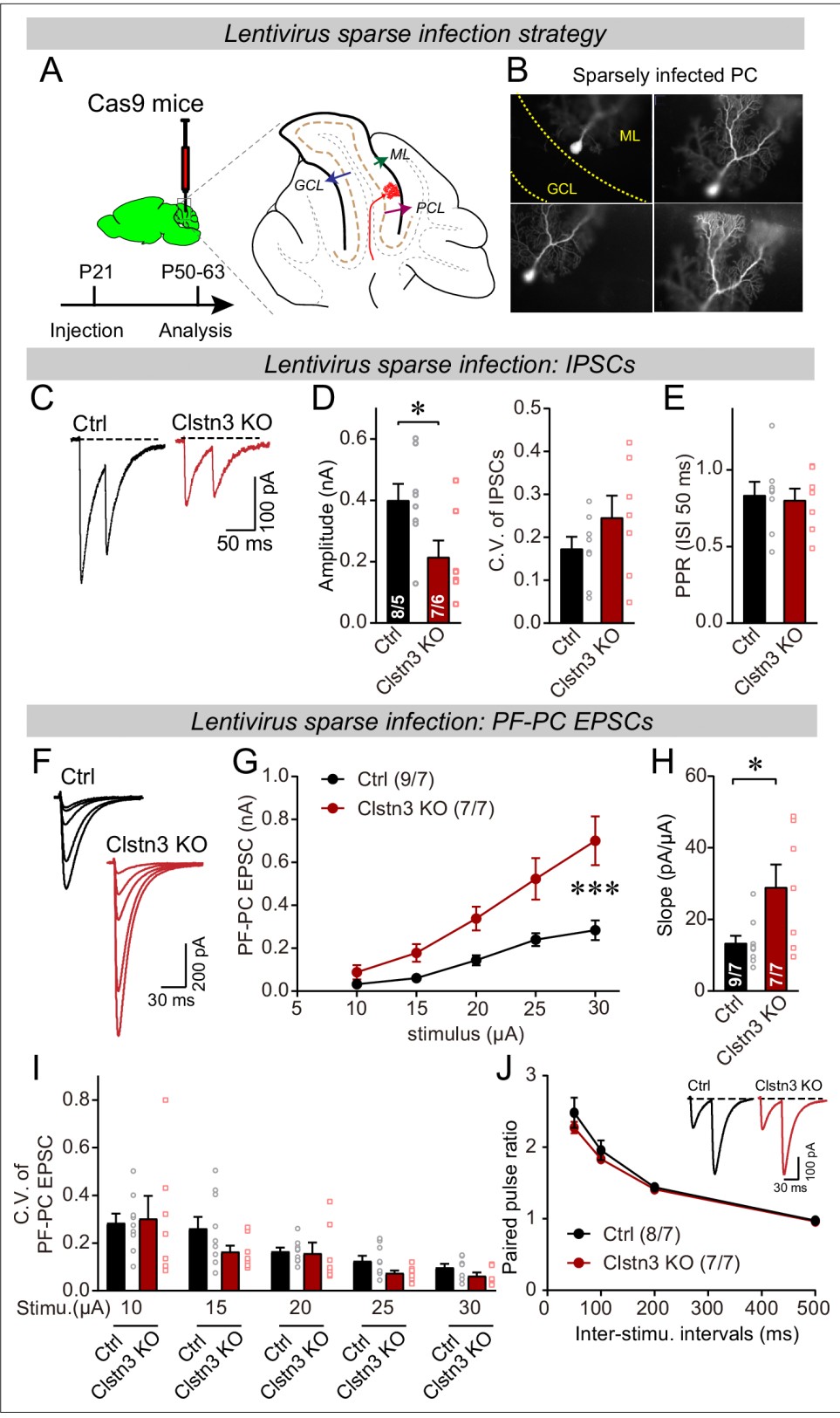

**Figure 11.** Sparse deletion of *Clstn3* in Purkinje cells by low-titer lentiviral infection recapitulates the phenotype obtained with the AAV-mediated deletion of *Clstn3* in the cerebellar cortex. (**A & B**) Strategy and validation of the sparse *Clstn3* deletion in Purkinje cells (**A**, experimental strategy; **B**, representative images of sparsely infected Purkinje cells, demonstrating that only isolated Purkinje cells and no granule or basket cells were infected).

*Figure 11 continued on next page*

*Figure 11 continued*

(**C–E**) The sparse *Clstn3* KO in Purkinje cells decreases the amplitude of evoked IPSCs in the cerebellar cortex without changing the release probability (**C**, representative IPSC traces; **D**, IPSC amplitudes (left) and coefficient of variation (right); **E**, paired-pulse ratio of IPSCs with a 50ms inter-stimulus interval). (**F–J**) The sparse *Clstn3* KO in Purkinje cells robustly increases the strength of parallel-fiber synapses, again without changing their release probability (**F**, representative traces; **G**, input/output curve of parallel-fiber EPSCs; **H**, summary graph of the slope of the input/output curve determined in individual cells; **I** coefficient of variation of evoked EPSCs at different stimulus intensities; **J**, plot of the paired-pulse ratio of parallel-fiber EPSCs). Data are means ± SEM. Two tailed unpaired t tests were applied to detect statistical significance with panels D, E and H. *p < 0.05. For panels G, I, and J, two-way ANOVA followed by post-hoc Tukey test was employed. For G, $F_{(4, 68)} = 31.695$, **p < 0.000.

The online version of this article includes the following figure supplement(s) for figure 11:

**Figure supplement 1.** CRISPR-mediated sparse *Clstn3* deletion in Purkinje cells causes a selective suppression of *Clstn3* expression, but not of *Clstn1* or *Clstn2* expression, in Purkinje cells, but not in basket or granule cells, as analyzed by single-cell quantitative reverse transcription-PCR (qRT-PCR).

for AAVs, and -as always- the experimenter was blinded to the identity of the viruses used. Imaging of cerebellar sections revealed that this approach led to very sparse infections of Purkinje cells, with no more than 2–3 Purkinje cells infected per section (*Figure 11B*).

We first asked whether the sparse *Clstn3* deletion in Purkinje cells causes a selective decrease of *Clstn3* expression in Purkinje cells but not in surrounding basket or granule cells. To address this question, we patched infected Purkinje cells and surrounding neurons and measured *Clstn1*, *Clstn2*, and *Clstn3* mRNA levels by qRT-PCR (*Figure 11—figure supplement 1A*). The sparse deletion of *Clstn3* only suppressed *Clstn3* levels but not *Clstn1* or *Clstn2* levels in Purkinje cells, and only suppressed *Clstn3* levels in Purkinje cells but not in surrounding granule or basket cells (*Figure 11—figure supplement 1B*). Thus, this approach enables specific suppression of *Clstn3* expression in sparsely infected Purkinje cells.

We then measured the effect of the sparse *Clstn3* deletion in Purkinje cells on excitatory and inhibitory synaptic responses. Strikingly, the sparse *Clstn3* deletion suppressed the strength of inhibitory synapses in Purkinje cells similar to the *Clstn3* deletion in the entire cerebellar cortex (~60% decrease; *Figure 11C–E*). At the same time and in the same type of neurons, the sparse *Clstn3* deletion increased the strength of parallel-fiber synapses (~100% increase; *Figure 11F–H*). Both the decrease in inhibitory synaptic strength and the increase in excitatory parallel-fiber synapse strength were not associated with significant alterations in the coefficient of variation or paired-pulse ratio of synaptic responses (*Figure 11D, E, I and J*), which is consistent with the changes in synapse numbers observed morphologically after pan-cerebellar *Clstn3* deletions (*Figures 4, 7 and 8*). Thus, the Clstn3 selective deletion in Purkinje cells produces both the increase in parallel-fiber synapses and the decrease in inhibitory synapses.

## Discussion

Calsyntenins are intriguing but enigmatic cadherins. Two distinct, non-overlapping roles were proposed: as a postsynaptic adhesion molecule promoting synapse formation, or as a presynaptic kinesin-adaptor protein mediating axonal transport of APP and other cargoes (*Araki et al., 2007*; *Kim et al., 2020*; *Konecna et al., 2006*; *Lipina et al., 2016*; *Pettem et al., 2013*; *Ster et al., 2014*; *Vagnoni et al., 2012*). Both functions are plausibly supported by extensive data, but neither function was conclusively tested. Here, we examined the role of one particular calsyntenin isoform, *Clstn3*, in one particular neuron, Purkinje cells that predominantly express this isoform. Our data establish that *Clstn3* acts as a postsynaptic adhesion molecule in Purkinje cells that is selectively essential for regulating synapse numbers, confirming an essential function for *Clstn3* as a postsynaptic adhesion molecule. Our data are surprising, however, in revealing that *Clstn3* functions not by universally promoting synapse formation, but by exerting opposite effects in different types of synapses in the same neurons. Specifically, our results demonstrate that the deletion of *Clstn3* causes a decrease in inhibitory basket- and stellate-cell synapses on Purkinje cells, but an increase in excitatory parallel-fiber synapses (*Appendix 1—figure 1*). Thus, *Clstn3* does not function simply as a synaptogenic

adhesion molecule, but acts as a regulator of the balance of excitatory and inhibitory synaptic inputs on Purkinje cells.

The functions we describe here for *Clstn3* are different from those of previously studied synaptic adhesion molecules or synapse-organizing signals. Whereas presynaptic adhesion molecules generally act in both excitatory and inhibitory synapses, few postsynaptic adhesion molecules were found to function in both. In the rare instances in which an adhesion molecule was documented to mediate signaling in excitatory and inhibitory synapses, such as is the case for *Nlgn3* (but not of other neuroligins), the adhesion molecule acts to promote synaptic function in both (*Chanda et al., 2017*; *Zhang et al., 2015*). Not only do we find that *Clstn3*, different from previously identified synaptic adhesion molecules, restricts the numbers of a specific synapse (parallel-fiber synapses), but also that *Clstn3* increases the numbers of another specific synapse (GABAergic basket- and stellate-cell synapses) in the same neurons.

The evidence supporting our conclusions can be summarized as follows. First, we showed that in the cerebellum, Purkinje cells express *Clstn3* at much higher levels than *Clstn1* or *Clstn2*, whereas other cerebellar neurons predominantly express *Clstn1* and exhibit only low levels of *Clstn3* (*Figure 1*, *Figure 1—figure supplement 1*). Second, we demonstrated efficient CRISPR-mediated deletion of *Clstn3* expression in Purkinje cells of the cerebellar cortex using viral tools (*Figure 2*, *Figure 2—figure supplements 1 and 2*), and showed that this deletion caused significant motor learning impairments (*Figure 3*). Third, we used quantitative immunocytochemistry with both pre- and postsynaptic markers to show that the cerebellar deletion of *Clstn3* decreases inhibitory synapses, increases excitatory parallel-fiber synapses, and leaves excitatory climbing-fiber synapses unchanged (*Figures 4 and 7*). Fourth, we reconstructed entire Purkinje cells using confocal microscopy, demonstrating a selective increase in spine numbers (*Figure 8*). Fifth, we employed extensive electrophysiological studies to document that the cerebellar *Clstn3* deletion suppresses the strength of inhibitory inputs on Purkinje cells, but elevates that of excitatory parallel-fiber inputs (*Figures 5, 6, 9 and 10*). Finally, we showed that the sparse deletion of *Clstn3* in Purkinje cells effectively and selectively suppresses *Clstn3* mRNA levels and replicates the electrophysiological phenotype of the pan-cerebellar deletion, consistent with the selective expression of high levels of *Clstn3* in Purkinje cells (*Figure 11*, *Figure 11—figure supplement 1*). These data, viewed together, document that *Clstn3* in cerebellum acts as a postsynaptic adhesion molecule that simultaneously enhances inhibitory synaptic inputs and suppresses parallel-fiber excitatory synaptic inputs. Particularly striking is the unexpected increase in parallel-fiber synapse numbers we observe. This increase was documented by six independent types of evidence: Spine counts in reconstructed neurons, the mEPSC frequency, evoked parallel-fiber EPSC amplitudes, and immunocytochemical analyses for pre- and postsynaptic markers and for a glial tripartite synapse marker. Each of these types of evidence is by itself incomplete but together they provide strong support for the overall conclusion that the deletion of Clstn3, an adhesion molecule, enhances parallel-fiber synapse numbers.

Several questions arise. First, why are the phenotypes we observe in *Clstn3* KO Purkinje cells so much stronger than those previously detected in CA1-region pyramidal neurons (*Kim et al., 2020*; *Pettem et al., 2013*)? This difference could be due to differences in cell type or to the more acute nature of our manipulations. More likely, however, this difference is caused by the lack of redundancy of *Clstn3* function in Purkinje cells, since other calsyntenin isoforms are co-expressed with *Clstn3* in CA1-region neurons at high levels (*Figure 1A*), but not in Purkinje cells (*Figure 1B–E*).

Second, what is the mechanism of *Clstn3* action at synapses? We used manipulations in young adult mice in which cerebellar synapses have been established but likely continuously turn over (*Attardo et al., 2015*; *Pfeiffer et al., 2018*). Because of this turnover, our data do not reveal whether *Clstn3* acts in the initial establishment and/or the maintenance of synapses, a somewhat artificial distinction since synapse formation may consist in the stabilization of promiscuous contacts and since synapses turn over continuously (*Südhof, 2021*). The functional consequences of these actions for cerebellar circuits are identical, in that they lead to a dramatic shift in excitatory/inhibitory balance in the cerebellar cortex.

Third, what trans-synaptic interactions mediate the functions of *Clstn3*? Several papers describe binding of calsyntenins to neurexins (*Kim et al., 2020*; *Pettem et al., 2013*). However, our data uncover a phenotype that is different from that observed with deletions of neurexins or neurexin ligands, suggesting that *Clstn3* may not exclusively function by binding to neurexins. The deletion of

the neurexin ligand Cbln1 leads to a loss of parallel-fiber synapses in the cerebellar cortex instead of a gain, suggesting that a different calsyntenin ligand is involved. Moreover, the specific conclusions of the papers describing calsyntenin-binding to neurexins differ (*Kim et al., 2020*; *Pettem et al., 2013*), leaving the interaction mode undefined. Thus, at present the most parsimonious hypothesis is that postsynaptic calsyntenins act by binding, at least in part, to presynaptic ligands that may include neurexins but likely also involve other proteins.

Fourth, does *Clstn3* physiologically act to restrict the formation of excitatory parallel-fiber synapses, leading to an increase in parallel-fiber synapses upon deletion of *Clstn3*, or is this increase an indirect compensatory effect produced by the decrease in inhibitory synapses? Multiple arguments support a specific action of *Clstn3* at parallel-fiber synapses. *Clstn3* protein was localized to parallel-fiber synapses by immunoelectron microscopy (*Hintsch et al., 2002*). Moreover, other genetic manipulations that cause a decrease in inhibitory synaptic transmission in cerebellar cortex, such as deletions of *Nlgn2* or of GABA$_A$-receptors (*Briatore et al., 2020*; *Fritschy et al., 2006*; *Meng et al., 2019*; *Zhang et al., 2015*), do not induce an increase in excitatory parallel-fiber synapses. Finally and probably most importantly, no competition between GABAergic and glutamatergic synapses has been observed, such that the decrease in one would lead to the increase of the other, even though competition between synapses using the same transmitters is well-described (e.g. competition between glutamatergic parallel- and climbing-fiber synapses on Purkinje cells; *Cesa and Strata, 2009*; *Miyazaki et al., 2012*; *Strata et al., 1997*). Quite the contrary, the rules of homeostatic plasticity would predict that a decrease in GABAergic synapses should lead to a decrease, not an increase, in glutamatergic synapses (*Monday et al., 2018*; *Nelson and Valakh, 2015*). Thus, our data overall suggest that *Clstn3* specifically acts to limit the formation of parallel-fiber synapses and enhance the formation of inhibitory synapses in the cerebellar cortex.

Our study also has clear limitations. We did not examine axonal or dendritic transport, and do not exclude the possibility that *Clstn3* performs additional functions as an adaptor for kinesin-mediated transport. Moreover, we did not address the possibility that different calsyntenins perform distinct functions, since we analyzed only a single isoform whose deletion produces a large phenotype. Furthermore, our data do not rule out the possibility that Clstn3 expression in other cerebellar cell types besides Purkinje cells has a functional role, which may contribute to the behavioral phenotypes we observed. In addition, our data do not inform us about the mechanism of action of Clstn3, which may act by a neurexin-dependent mechanism. The example of neuroligins shows that a synaptic adhesion molecule can have both a neurexin-dependent and neurexin-independent functions (*Ko et al., 2009*; *Wu et al., 2019*). Finally, we cannot exclude the possibility that Clstn3 has additional functions in Purkinje cells that were not detected in our experiments, either because the resulting phenotypes are too subtle or because the additional functions are redundant with those of other calsyntenins expressed at low levels. For example, it is possible that low levels of Clstn3 are present in climbing-fiber synapses and/or that they are functionally redundant with Clstn1 or Clstn2 in these synapses; in both cases, our analyses would miss an additional function for Clstn3 in climbing-fiber synapses.

Multiple synaptic adhesion molecules have already been implicated in synapse formation in Purkinje cells. The interaction of presynaptic neurexins with cerebellins and postsynaptic GluD receptors plays a major role in shaping parallel-fiber synapses (*Yuzaki and Aricescu, 2017*), and the binding of C1ql1 to postsynaptic Bai3 mediates climbing-fiber synapse formation (*Kakegawa et al., 2015*; *Sigoillot et al., 2015*). Postsynaptic *Nlgn2* and *Nlgn3* are major contributors to the function of GABAergic synapses in Purkinje cells (*Zhang et al., 2015*), as is dystroglycan (*Briatore et al., 2020*). How do various synaptic adhesion complexes collaborate in establishing and shaping different types of synapses on Purkinje cells? Do these molecules act sequentially at different stages, or work in parallel? The overall view of synapse formation that emerges from these studies resembles that of a baroque orchestra lacking a conductor, in which different players individually contribute distinct but essential facets to the work that is being performed. In this type of orchestra, some players, such as neurexins, play prominent roles in coordinating the actions of their sections, whereas others, such as latrophilins, initiate movements. According to this scenario, *Clstn3* (and possibly other calsyntenins) may regulate the loudness of different sections of the orchestra, or translated into the terms of a synapse, control the efficacy of signals regulating excitatory vs. inhibitory synapses. However, it is also possible that *Clstn3* acts directly in the process of synapse formation itself, but differentially affects distinct synapses depending on the presence of different ligands – time will tell!

# Materials and methods

## Key resources table

| Reagent type (species) or resource | Designation | Source or reference | Identifiers | Additional information |
|---|---|---|---|---|
| Genetic reagent (*Mus musculus*) | Constitutive Cas9 | PMID:25263330 | JAX ID: 024858 | |
| Genetic reagent (*Mus musculus*) | Pcp2-Cre | PMID:11105049 | JAX ID: 004146 | |
| Genetic reagent (*Mus musculus*) | RiboTag | PMID:19666516 | JAX ID: 029977 | |
| Cell line (*Homo sapiens*) | HEK293T | ATCC | CRL-11268 | |
| Recombinant DNA reagent | AAV-U6-sg66-U6-sg21-CAG tdTomato | This paper | | Sg66 and sg21 were cloned in an AAV backbone and made into the AAVDJ serotype. See sgRNA design and generation of Vectors for cloning details, Virus production for how the AAVs were produced |
| Recombinant DNA reagent | Lentiviral sg66 and sg21-CAG tdTomato | This paper | | Sg66 and sg21 were cloned into a lentiviral shuttle plasmid for sparse infection. See Virus production for cloning details and how lentiviruses were produced. |
| Sequence-based reagent | *Clstn1* RNA FISH probe | Advanced Cell Diagnostics | Cat: 542611 | |
| Sequence-based reagent | *Clstn2* RNA FISH probe | Advanced Cell Diagnostics | Cat: 542621 | |
| Sequence-based reagent | *Clstn3* RNA FISH probe | Advanced Cell Diagnostics | Cat: 542631 | |
| Sequence-based reagent | *Clstn3* qPCR primers and probe | This paper | | See Quantitative RT-PCR and *Figure 2—figure supplement 1A* for how primers and probe were designed |
| Antibody | Anti-Clstn3 (rabbit polyclonal) | PMID:24613359 | | Primary antibody, (1:1000) IB |
| Antibody | Anti-Actb (mouse monoclonal) | Sigma | #A1978 | Primary antibody, (1:10000) IB |
| Antibody | Anti-vGluT1 (rabbit polyclonal) | Yenzym | YZ6089 | Primary antibody, (1:1000) IHC |
| Antibody | Anti-vGluT2 (rabbit polyclonal) | Yenzym | YZ6097 | Primary antibody, (1:1000) IHC |
| Antibody | Anti-vGAT (guinea pig polyclonal) | Sysy | 131004 | Primary antibody, (1:1000) IHC |
| Antibody | Anti-GluA1 (rabbit polyclonal) | Millipore | AB1504 | Primary antibody, (1:1000) IHC |
| Antibody | Anti-GluA2 (mouse monoclonal) | Millipore | MAB397 | Primary antibody, (1:1000) IHC |
| Antibody | Anti-GABA$_{(A)}$Rα1 (mouse monoclonal) | Neuromab | N95/35 | Primary antibody, (1:1000) IHC |
| Chemical compound, drug | Tribromoethanol | Sigma | T48402 | 250 mg/kg for anesthesia |
| Chemical compound, drug | Picrotoxin | Tocris | 1128 | |
| Chemical compound, drug | APV | Tocris | 0106 | |
| Chemical compound, drug | CNQX | Tocris | 1045 | |
| Chemical compound, drug | NBQX | Tocris | 1044 | |
| Chemical compound, drug | QX314 | Tocris | 1014 | |

*Continued on next page*

*Continued*

| Reagent type (species) or resource | Designation | Source or reference | Identifiers | Additional information |
|---|---|---|---|---|
| Chemical compound, drug | Tetrodotoxin | Cayman Chemical | 14964 | |
| Chemical compound, drug | DAPI | Sigma | D8417 | |
| Chemical compound, drug | Biocytin | Sigma | B4261 | |
| Chemical compund, drug | Pepsin | DAKO | S3002 | 1 mg/ml |
| Sequence-based reagent | *Clstn1* | IDT | Mm.PT.58.6236597 | commercially designed |
| Sequence-based reagent | *Clstn2* | IDT | Mm.PT.58.6443231 | commercially designed |
| Sequence-based reagent | *Clstn3* | IDT | Mm.PT.58.45847813.g | commercially designed |
| Sequence-based reagent | *Gapdh* | Applied Biosystems | 4352932E | commercially designed |
| Software, algorithm | SnapGene | GSL Biotech | | previously existing |
| Software, algorithm | Image Studio Lite | LI-COR | | previously existing |
| Software, algorithm | pClamp10 | Molecular Device | | previously existing |
| Software, algorithm | Clampfit10 | Molecular Device | | previously existing |
| Software, algorithm | NIS-Elements AR | Nikon | | previously existing |
| Software, algorithm | ImageJ | NIH | | previously existing |
| Software, algorithm | Neurolucida360 | MBF science | | previously existing |
| Software, algorithm | Adobe Illustrator | Adobe | | previously existing |
| Software, algorithm | Graphpad Prism 8.0 | Graphpad software | | previously existing |

IB: immunoblotting, IHC: immunohistochemistry

## Animals

Constitutive Cas9 mice (https://www.jax.org/strain/024858) were used and maintained as homozygotes (*Platt et al., 2014*). Pcp2-Cre mice were crossed with RiboTag mice to obtain Pcp2-RiboTag mice, which enabled specific HA tagging of ribosomes in cerebellar purkinje cells. Analyses were performed on littermate mice. Mice were fed ad libitum and maintained on 12 hr light dark cycles. All animal experiments: All protocols were carried out under the National Institutes of Health Guidelines for the Care and Use of Laboratory Animals and were approved by the Administrative Panel on Laboratory Animal Care (APLAC) at Stanford University and institutional animal care and use committee (IACUC). The animal protocol #20787 was approved by Stanford University APLAC and IACUC. All surgeries were performed under avertin anesthesia and carprofen analgesia, and every effort was made to minimize suffering, pain, and distress.

Single-molecule RNA fluorescent in situ hybridization (smRNA-FISH) smRNA-FISH in-situ hybridization experiment was performed on brain sections from P30 wild type C57BL/6 J mice according to the manufacturer instructions using Multiplex Fluorescent Detection Reagents V2 kit (# 323110, Advanced Cell Diagnostics). Predesigned probes for Clstn1 (# 542611), Clstn2 (# 542621), and Clstn3 (# 542631) were purchased from ACD. sgRNA design and generation of Vectors sgRNAs were designed using protocols developed by the Zhang lab (https://zlab.bio/guide-design-resources) to minimize potential off-target effects. The pAAV construct was modified from Addgene #60231 (*Platt et al., 2014*) with Cre-GFP replaced by tdTomato and human synapsin by CAG promoter to allow efficient expression in the cerebellum. Two sgRNAs were cloned in a single vector using Golden Gate Cloning assembly. Empty vector without sgRNAs was used as control. Genome editing efficiency of sgRNAs and potential off-target editing effects were initially evaluated according to previous report (*Brinkman et al., 2014*; *Brinkman and Van Steensel, 2019*). Briefly, forward and reverse primers targeted genomic DNA were designed to flank the potential sgRNA editing sites and off-target

editing sites (*Figure 2—figure supplement 1*). PCR products (300–1000 bp) from control and KO groups were sequenced and compared on TIDE website (https://tide.nki.nl/) (*Brinkman et al., 2014*; *Brinkman and Van Steensel, 2019*).

Primers for sg66 and sg21 editing site sequencing:

| sgRNA | Forward primer | Reverse primer |
|---|---|---|
| sg66 | AGTAGTCCCTTCCCCACAGG | GATGTGAGGACCCCATGACC |
| sg21 | GTGTGAGGAGGAGAATGGGC | AGGCAAAGTGGGGTGAGATG |

Primers for off-target site sequencing on sg66:

| Chromosome | Forward primer | Reverse primer |
|---|---|---|
| chr7 | GAACCCCAAGTACGCCAAGA | TTGACAGTGTGTGGCTGTGT |
| chr2 | TGCTCCGAGGTCTCCCTAAA | AAGGTTCCAGGTCCTGTTGC |
| chr13 | AAGAGATCCCTCCGAACATGG | GCCCATCTGACAGGAGTATGT |

Primers for off-target site sequencing on sg21:

| Chromosome | Forward primer | Reverse primer |
|---|---|---|
| chr17 | GGCAGATCTCTCGTGATGGC | TTAGTCTTGGCTGCGTCACC |
| chr5 | GGAACAAAAAGCCTGGCTCC | AATCTGGGCTGGCTCATTCC |
| chr13 | AGAGAAGGGAATGGGACCGA | ATGGCTCAGCGATTAGTGGG |

## Virus preparation and stereotactic injections

AAV: pAAV carrying sgRNAs was serotyped with the AAV-DJ capsid (*Grimm et al., 2008*). Briefly, helper plasmids (phelper and pDJ) and AAV-sgRNA vector were co-transfected into HEK293T cells (ATCC, CRL-11268), at 4 µg of each plasmid per 30 cm$^2$ culture area, using the calcium phosphate method. Cells were harvested 72 hr post-transfection, and nuclei were lysed and AAVs were extracted using a discontinuous iodixanol gradient media at 65,000 rpm for 3 hr. AAVs were then washed and dialyzed in DMEM and stored at –80 °C until use. Genomic titer was tested with qPCR and adjusted to 5 × 10$^{12}$ particles/ml for in vivo injections.

Lentivirus: U6-sgRNAs or controls, and CAG-tdTomato were cloned into the lentiviral backbone FUW and lentiviruses were produced with three helper plasmids (pRSV-REV, pRRE, and VSVG expression vector) in HEKT293 cells (ATCC) using calcium phosphate, at 5 µg of each plasmid per 25 cm$^2$ dish area, respectively. Media with viruses was collected at 36–48 hr after transfection, centrifuged at 3000 x g for 20 min to remove debris, filtered at 0.45 µm, and ultracentrifuged at 55,000 x g for 2 hr. Pellets were re-suspended with 300 µl DMEM for every 30 ml of the initial volume to achieve a lower titer. Viruses were aliquoted and stored at –80 °C until further use.

Stereotactic injection: p21 Cas9 mice were anesthetized with tribromoethanol (250 mg/kg, T48402, Sigma, USA), head-fixed with a stereotaxic device (KOPF model 1900). AAVs carrying sgRNAs or control viruses were loaded via a glass pipette connected with a 10 µl Hamilton syringe (Hamilton, 80308, US) on a syringe injection pump (WPI, SP101I, US) and injected at a speed of 0.15 µl/min. Pipette was left in cerebellum for additional 5 min after injection completion. Carprofen (5 mg/kg) was injected subcutaneously as anti-analgesic treatment. To infect the whole cerebellum, we injected multiple sites evenly distributing over the cerebellum skull, coordinates were as previously reported (*Zhou et al., 2020*), anterior to bregma, lateral to midline, ventral to dura (mm): (–5.8, ± 0.75), (–5.8, ± 2.25), (–6.35, 0), (–6.35, ± 1.5), (–6.35, ± 3), (–7, ± 0.75), and (–7, ± 2.25), with a series of depth (mm): 2, 1.5, 1, 0.5, and volume was 0.25 µl/depth. To achieve sparse infection with lentiviruses, we injected at (–6.35, 0) with a series of depth (mm) 2, 1.5, 1, 0.5 with a volume of 0.25 µl/ depth. All viruses were coded during virus injection and remained blinded throughout the whole study until data analyses were done.

## Quantitative RT-PCR

Virus-infected cerebellar tissue indicated by tdTomato was carefully dissected under fluorescence microscope. RNA was extracted using Qiagen RNeasy Plus Mini Kit with the manufacturer's protocol

(Qiagen, Hilden, Germany). Quantitative RT-PCR was run in QuantStudio 3 (Applied biosystems, Thermo Fisher Scientific, USA) using TaqMan Fast Virus 1-Step Master Mix (PN4453800, Applied biosystems, Thermo Fisher Scientific, USA). PrimerTime primers and FAM-dye coupled detection probes were used for detecting *Clstn3* mRNA level. To detect genome editing efficiency, qPCR primers and probe were targeting the two exons and designed to flank the double-strand breaks of the two sgRNAs (*Yu et al., 2014*). (*Clstn3*: Forward primer: AGAGTACCAGGGCATTGTCA; reverse primer: GATCACAGCCTCGAAGGGTA; probe: TGGATAAAGATGCTCCACTGCGCT, also see *Figure 2— figure supplement 1A*). A commercially available *Gapdh* probe was used as internal control (Cat: 4352932E, Applied Biosystems).

## Immunohistochemistry

Immunohistochemistry on the cerebellar cortex was done as previously reported (*Zhang et al., 2015*). Mice were anesthetized with isoflurane and sequentially perfused with phosphate buffered saline (PBS) and ice cold 4% paraformaldehyde (PFA). Brains were dissected and post-fixed in 4% PFA for 15 min, then cryoprotected in 30% sucrose in PBS for 24 hr. Forty-µm-thick sagittal sections of cerebellum were collected using a Leica CM3050-S cryostat (Leica, Germany). Free floating brain sections were incubated with blocking buffer (5% goat serum, 0.3% Triton X-100) for 1 hr at room temperature, then treated with primary antibodies diluted in blocking buffer overnight at 4 °C (anti-vGluT1, Rabbit, YZ6089, Yenzym, 1:1,000; anti-vGluT2, Rabbit, YZ6097, 1:1,000, Yenzym; anti-vGAT, guinea pig, 131004, Sysy,1:1,000; anti-GluA1, Rabbit, AB1504, 1:1000; anti-GluA2, mouse, MAB397, 1:1000). Sections were washed three times with PBS (15 min each), then treated with secondary antibodies (Alexa goat anti guinea pig 633, A-21105, Invitrogen, 1:1,000; or Alexa goat anti rabbit 647, A-21245, Invitrogen, 1:1,000) for 2 hr at room temperature. After washing with PBS 4 times (15 min each), sections were stained with DAPI (D8417, Sigma) and mounted onto Superfrost Plus slides with mounting media. Confocal images were acquired with a Nikon confocal microscope (A1Rsi, Nikon, Japan) with 60 x oil objective, at 1024 × 1024 pixels, with z-stack distance of 0.3 µm. All acquisition parameters were kept constant within the same day between control and *Clstn3* KO groups. Images were taken from cerebellar lobules IV/V. Images were analyzed with Nikon analysis software. During analysis, we divided the cerebellar cortex into different layers to compare *Clstn3* KO effects. We defined 0–40% as superficial molecular layer and 40–80% as molecular deep layers, 80–100% as PCL, and we analyzed and labeled GCL separately in vGAT staining.

For GABA$_{(A)}$Rα1 subunit antibody staining, antigen retrieval was achieved via pepsin digestion according to antibody instruction and as previously described (*Franciosi et al., 2007*). Briefly, free-floating sagittal cerebellum slices were incubated in PBS at 37 °C for 5 min, then washed in distilled water, and transferred to 1 mg/ml pepsin at 37 °C for another 2 min. After washing in PBS for 5 min x 3 times, slices were undergoing immunohistochemistry as described above.

## Immunoblotting

Immunoblotting was performed as described previously (*Zhang et al., 2015*). Mice were anesthetized with isoflurane and decapitated on ice, with the cerebellum dissected out and homogenized in RIPA buffer (in mM: 50 Tris-HCl pH7.4, 150 NaCl, 1% Triton X-100, 0.1% SDS, 1 EDTA) with protease inhibitor cocktail (5056489001, Millipore Sigma) and kept on ice for 30 min. Samples were centrifuged at 14,000 rpm for 20 min at 4 °C, supernatant were kept and stored in –80 °C until use. Proteins were loaded onto 4%–20% MIDI Criterion TGX precast SDS-PAGE gels (5671094, Bio-Rad), and gels were blotted onto nitrocellulose membranes using the Trans-blot turbo transfer system (Bio-Rad). Membranes were blocked in 5% milk diluted in PBS for 1 hr at room temperature, then incubated overnight at 4°C with primary antibodies diluted in 5% milk in TBST (0.1% Tween-20). Primary antibodies of anti-Clstn3 (Rabbit, 1:1000) was previously described (*Um et al., 2014*). Antibody against beta-actin from Sigma (A1978, Mouse, 1:10,000) was used as a loading control.

Membranes were then washed with TBST and incubated with fluorescence labeled IRDye secondary antibodies (IRDye 680LT Donkey anti-Rabbit, 926–68023, LI-COR, 1:10,000; IRDye 800CW donkey anti-mouse, 926–68023, LI-COR, 1:10,000). Signals were detected with Odyssey CLx imaging systems (LI-COR) and data were analyzed with Image Studio 5.2 software. Total intensity values were normalized to actin prior to control.

## Electrophysiology

Cerebellar electrophysiology was carried out as described previously (*Caillard et al., 2000*; *Foster and Regehr, 2004*; *Llano et al., 1991*; *Zhang et al., 2015*). Briefly, the cerebellum was rapidly removed and transferred into continuously oxygenated ice cold cutting solutions (in mM: 125 NaCl, 2.5 KCl, 3 $MgCl_2$, 0.1 $CaCl_2$, 25 glucose, 1.25 $NaH_2PO_4$, 0.4 ascorbic acid, 3 myo-inositol, 2 Na-pyruvate, and 25 $NaHCO_3$). 250 µm sagittal slices were cut using a vibratome (VT1200S, Leica, Germany) and recovered at room temperature for >1 hr before recording. Oxygenated ACSF (in mM: 125 NaCl, 2.5 KCl, 1 $MgCl_2$, 2 $CaCl_2$, 25 glucose, 1.25 $NaH_2PO_4$, 0.4 ascorbic acid, 3 myo-inositol, 2 Na-pyruvate, and 25 $NaHCO_3$) was perfused at 1 ml/min during recording. Whole cell recordings with Purkinje cells were from cerebellar lobules IV/V, with patch pipettes (2–3 MΩ) pulled from borosilicate pipettes (TW150-4, WPI, USA) using PC-10 puller (Narishige, Japan). The following internal solutions were used (in mM): (1) for EPSC, 140 $CsMeSO_3$, 8 CsCl, 10 HEPES, 0.25 EGTA, 2 Mg-ATP, 0.2 Na-GTP (pH adjusted to 7.25 with CsOH); (2) for IPSC, 145 CsCl, 10 HEPES, 2 $MgCl_2$, 0.5 EGTA, 2 Mg-ATP, 0.2 Na-GTP (pH adjusted to 7.25 with CsOH). Liquid junction was not corrected during all recordings. For all EPSC recordings, 50 µM picrotoxin (1128, Tocris) and 10 µM APV (0106, Tocris) were contained in ACSF, and (1) additionally 0.5 µM NBQX (1044, Tocris) were included for climbing-fiber EPSC recordings; (2) 1 µM TTX (Tetrodotoxin, 14964, Cayman Chemical) for mEPSC recordings. For all IPSC recordings, (1) 10 µM CNQX (1045, Tocris) and 10 µM APV were included in ACSF, and (2) 1 µM TTX were included in mIPSC recordings. A glass theta electrode (64–0801, Warner Instruments, USA, pulled with PC-10) filled with ACSF was used as stimulating electrode, and littermate control and Clstn3 knockout mice were analyzed meanwhile using the same stimulating electrode. For climbing fibers, the electrode was placed in the granule cell layer around Purkinje cells and identified by all-or-none response, together with paired-pulse depression at 50ms inter-stimulus interval (*Eccles et al., 1966*; *Zhang et al., 2015*). For parallel fibers, electrodes were placed in distal molecular layer with the same distance for both control and Clstn3 knockout mice (~200 µm from the recorded Purkinje cell) and identified by paired-pulse facilitation at 50ms inter-stimulus interval (*Zhang et al., 2015*). For basket cell stimulations, electrodes were placed in the proximal molecular layer (within~100 µm from the recorded Purkinje cell) and also identified with all-or-none response (*Caillard et al., 2000*; *Zhang et al., 2015*).

Membrane resistance and capacitances were calculated according to Ohm's law. Briefly, under voltage-clamp mode, cells were clamped at –70 mV after whole-cell mode was achieved. Then a 5 mV hyperpolarizing pulse was injected into the cell and membrane resistance was calculated by the equation: R=ΔU/ΔI, where ΔU = 5 mV, and ΔI was the current change in response to it. Capacitance was calculated by the equation: C = $\tau$/R, where $\tau$ was the decay time constant of the current trace fitted by an exponential curve, and R was the membrane resistance. Junction potentials were not changed throughout the studies. Cells with >20% changes in series resistances were rejected for further analysis. All electrophysiological data were sampled with Digidata1440 (Molecular Device, USA) and analyzed with Clampfit10.4. Coefficient of variation was calculated according to previous report (*Lisman et al., 2007*).

## Biocytin labeling in Purkinje cells

Two mg/ml Biocytin (B4261, Sigma) was dissolved in Cs-methanesulfonate internal solution followed by whole-cell voltage clamp recordings in the Purkinje cells (*Sando et al., 2019*). Slices were fixed in 4% PFA/PBS solution overnight at 4 °C. Slices were then washed 3 × 5 min with PBS, permeabilized, and blocked in 5% goat serum, 0.5% Triton-X100 in PBS at room temperature for 1 hr. Then slices were incubated in 1:1000 diluted Streptavidin Fluor 647 conjugate (S21374, Invitrogen) at room temperature for 2 hr in 5% goat serum in PBS, washed 5 × 5 min with PBS, and mounted onto Superfrost Plus slides for imaging. Image overviews were obtained with a Nikon confocal microscope (A1Rsi, Nikon, Japan) with a 60 x oil objective, at 1024 × 1024 pixels, with z-stack distance of 2 µm. Dendritic tree 3D reconstructions were performed using Neurolucida360 software (MBF science, USA) in the Stanford Neuroscience Microscopy Service Center. Note that some somas could not be detected automatically and were manually labeled. Spine images were obtained with a ZESS LSM980 inverted confocal, Airyscan2 for fast super-resolution setup, equipped with an oil-immersion 63 X objective. Z-stacks were collected at 0.2 µm intervals at 0.06 µm/pixel resolution with Airyscan2. Spine images were deconvolved using ZEN blue software (ZESS). Spine density and characteristics were analyzed with

Neurolucida360 software (MBF science, USA) in Stanford Neuroscience Microscopy Service Center. Only last order dendrites were analyzed, with 5–8 dendrites per cell per layer.

## Sing-cell qPCR

Sing-cell qPCR experiments were performed as previously described (*Fuccillo et al., 2015*). Acute sagittal cerebellum brain slices were cut as described above. Patch-pipettes were utilized for cytosol extraction. Samples underwent further processing according to sing-cell qPCR kit instructions (Thermofisher, Cat no. 4458237). Pre-amplified cDNAs were then processed for real-time qPCR on QuantStudio 3 (Applied biosystems, Thermo Fisher Scientific, USA) using TaqMan Fast Virus 1-Step Master Mix (PN4453800, Applied biosystems, Thermo Fisher Scientific, USA). Primers and FAM-dye coupled detection probes of *Clstn1*, *Clstn2*, *Clstn3* were pre-designed and ordered from IDT (*Clstn1*, Mm.PT.58.6236597; *Clstn2*, Mm.PT.58.6443231; *Clstn3*, Mm.PT.58.45847813.g). A commercially available *Gapdh* probe was used as internal control (Cat: 4352932E, Applied Biosystems). To ensure the specificity and efficiency of amplification, all assays were tested with serial dilutions of mouse cerebellum cDNA to verify efficiency (90%–110%) and linear amplification ($R^2 > 0.96$). mRNA levels were calculated by normalizing Ct values of relative genes to *Gapdh*.

## Behavior

### Accelerating rotarod

Accelerating rotarod was carried out as previously reported (*Rothwell et al., 2014*). Mice were placed on an accelerating rotarod (IITC Life Science). The rod accelerated from 4 to 40 r.p.m. in 5 min. Mice were tested 3 times per day with 4 hr interval and repeated for 3 days. Time stayed on the rod was recorded while the mouse fell off, or hanged on without climbing, or reached 5 min.

### Three-chamber social interaction

Social interaction was evaluated in a three-chamber box. Mice were placed initially in the central chamber to allow 10 min habituation for all three chambers. For sociability session, a same sex- and age matched stranger mouse (stranger1) was placed inside an upside-down wire pencil cup in one of the side chambers. The other side had the same empty pencil cup. A test mouse was allowed 10 min to investigate the three chambers. For social novelty session, another stranger mouse (stranger2) was placed into the empty pencil cup and test mouse was allowed another 10 min to investigate between three chambers. The time mice spent in each chamber was recorded and analyzed using BIOBSERVE III tracking system.

### Open field

Mice were placed in the center of 40 × 40 cm white box and allowed to freely explore for 15 min. Videos were recorded and analyzed by BIOBSERVE III software. The 20 × 20 cm region in the center was defined as central zone. The total distance travelled and the activity exploring the center area were analyzed to evaluate the subject's locomotor ability and anxiety levels.

### Beam-walk

Beam-walk test was performed as previously described (*Burré et al., 2015*). Briefly, a cylindrical beam (80 cm in length, 1 cm or 2 cm in diameter) was fixed on both ends 40 cm above the table. A lamp was placed above the start point and serves as aversive stimulation. A small black box with nesting material is was placed at the end of the beam. During training days, mice were placed onto the start point of 2 cm diameter beam and were left to cross the beam to reach the cage. Mice were trained for a total three trials with 1 min interval. During the test days, 1 cm beam was used and the number of foot slips for each mouse per trial and the crossing time were recorded.

### Gait analysis

We performed gait analysis as described previously (*Wertman et al., 2019*; *Seigneur and Südhof, 2018*). Briefly, a 50 cm long, 10 cm wide tunnel was built 10 cm above the table. A piece of white paper with the same size was placed on the tunnel floor. Nesting materials from homecage were placed at the end. Nontoxic water-based paint was painted onto the forepaws (red) and hindpaws

(blue). Then the mice were allowed to walk through the tunnel to the box containing homecage bedding. Each mouse was tested three times. We measured stride length, stance, as well as the overlap between forepaw and hindpaw. Stride length was defined as the averaged distance between the successive footprints. Stride width was averaged distance between the left and right forepaws. Overlap was calculated as the averaged distance between forepaw and hindpaw footprint centers on the same side.

## Data analysis

Experiments and data analyses were performed blindly by coding viruses. Unpaired t-test or one-way ANOVA or two-way ANOVA or repeat measures ANOVA were used to analyze slice physiology data or immunohistochemistry data or behavior data as indicated in figure legends. Kolmogorov-Smirnov test was used to analyze the cumulative curves of mEPSCs or mIPSCs. Significance was indicated as *$p < 0.05$, **$p < 0.01$, ***$p < 0.001$. Data are expressed as means ± SEM.

## Acknowledgements

We thank Drs. Bo Zhang, Juliana Salgado, Justin Howard Trotter, Karthik Raju, Mu Zhou, and Richard Sando for advice, discussions, and help in this project. We thank Stanford Neuroscience Microscopy Center for the help with spine imaging. This paper was supported by a grant from NIH (MH052804 to TCS).

## Additional information

### Funding

| Funder | Grant reference number | Author |
|---|---|---|
| National Institute of Mental Health | MH052804 | Thomas C Südhof |

The funders had no role in study design, data collection and interpretation, or the decision to submit the work for publication.

### Author contributions

Zhihui Liu, Conceptualization, Data curation, Formal analysis, Investigation, Methodology, Resources, Validation, Visualization, Writing – original draft, Writing – review and editing; Man Jiang, Conceptualization, Data curation, Methodology, Resources, Validation, Visualization, Writing – review and editing; Kif Liakath-Ali, Data curation, Methodology; Alessandra Sclip, Roger Shen Zhang, Methodology; Jaewon Ko, Resources; Thomas C Südhof, Conceptualization, Formal analysis, Funding acquisition, Methodology, Project administration, Resources, Supervision, Writing – original draft, Writing – review and editing

### Author ORCIDs

Zhihui Liu ⓘ http://orcid.org/0000-0002-9198-9090
Man Jiang ⓘ http://orcid.org/0000-0002-0470-8722
Kif Liakath-Ali ⓘ http://orcid.org/0000-0001-9047-7424
Alessandra Sclip ⓘ http://orcid.org/0000-0002-9313-4176
Jaewon Ko ⓘ http://orcid.org/0000-0001-9184-1574
Thomas C Südhof ⓘ http://orcid.org/0000-0003-3361-9275

### Ethics

All animal experiments: All protocols were carried out under the National Institutes of Health Guidelines for the Care and Use of Laboratory Animals and were approved by the Administrative Panel on Laboratory Animal Care (APLAC) at Stanford University and institutional animal care and use committee (IACUC). The animal protocol #20787 was approved by Stanford University APLAC and IACUC. All surgeries were performed under avertin anesthesia and carprofen analgesia, and every effort was made to minimize suffering, pain, and distress.

## Decision letter and Author response

Decision letter https://doi.org/10.7554/eLife.70664.sa1
Author response https://doi.org/10.7554/eLife.70664.sa2

## Additional files

### Supplementary files
• Transparent reporting form

### Data availability

All data generated or analyzed during this study are included in the manuscript and supporting files. All numerical data have been provided in a zipped folder.

The following previously published dataset was used:

| Author(s) | Year | Dataset title | Dataset URL | Database and Identifier |
| --- | --- | --- | --- | --- |
| Saunders A, Macosko EZ, Wysoker A, Goldman M, Krienen F, de Rivera H, Bien E, Baum M, Wang S, Bortolin L, Goeva A, Nemesh J, Kamitaki N, Brumbaugh S, Kulp D, McCarroll SA | 2018 | Molecular Diversity and Specializations among the Cells of the Adult Mouse Brain | https://www.ncbi.nlm.nih.gov/geo/query/acc.cgi?acc=GSE1164 | NCBI Gene Expression Omnibus, GSE1164 |

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

## Appendix 1

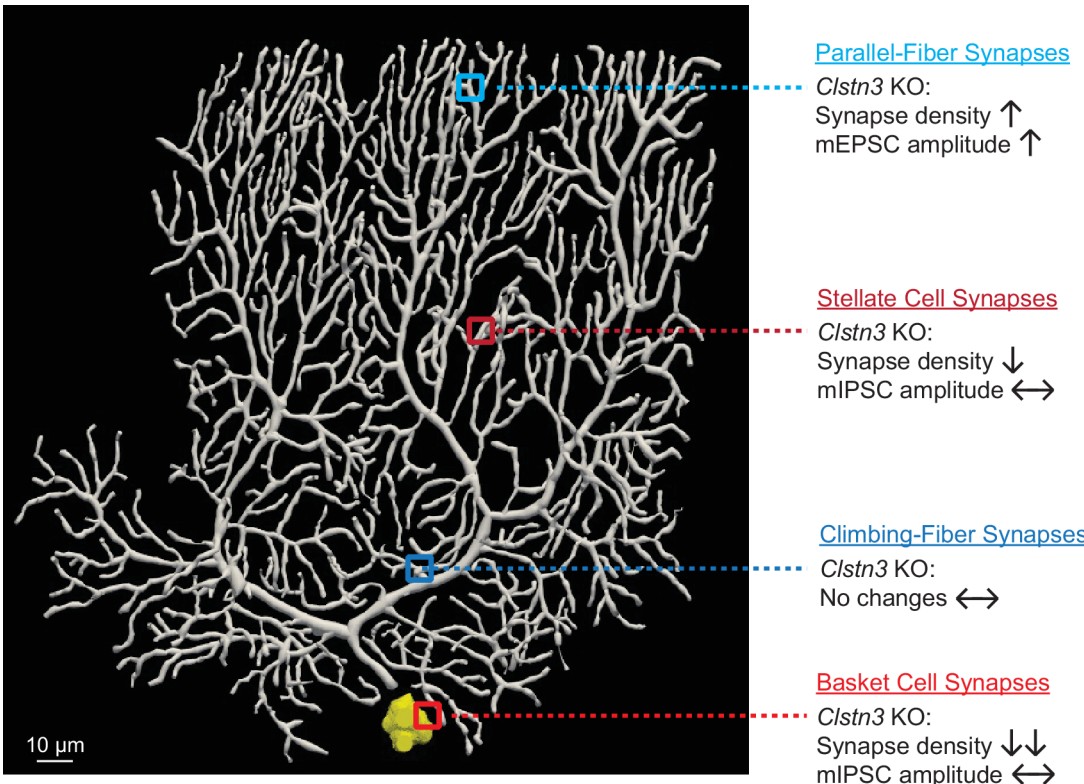

**Parallel-Fiber Synapses**
*Clstn3* KO:
Synapse density ↑
mEPSC amplitude ↑

**Stellate Cell Synapses**
*Clstn3* KO:
Synapse density ↓
mIPSC amplitude ⟷

**Climbing-Fiber Synapses**
*Clstn3* KO:
No changes ⟷

**Basket Cell Synapses**
*Clstn3* KO:
Synapse density ↓↓
mIPSC amplitude ⟷

**Appendix 1—figure 1.** Summary of synaptic changes induced in Purkinje cells by the *Clstn3* KO. The Purkinje cell image is from one of the cells reconstructed during the present study. The changes summarized on the right were identified in *Figures 3–10*.

