## [Editor Report]

This study investigated the role of calsyntenin-3, an atypical cadherin, in controlling synaptic inputs to cerebellar Purkinje cells. It provides compelling evidence that elimination of calsyntenin-3 from cells in the cerebellar cortex alters the E/I balance for Purkinje cells, although Purkinje cell-specific manipulations were used in only some of the experiments. The results indicated that calsyntenin-3 increases the strength of excitatory parallel fiber inputs and decreases the strength of inhibitory inputs.

---

## [Decision Letter]

**Decision letter after peer review:**

Thank you for submitting your article "Calsyntenin-3, an atypical cadherin, suppresses inhibitory basket- and stellate-cell synapses but boosts excitatory parallel-fiber synapses in cerebellum" for consideration by *eLife*. Your article has been reviewed by 3 peer reviewers, one of whom is a member of our Board of Reviewing Editors, and the evaluation has been overseen by a Reviewing Editor and Gary Westbrook as the Senior Editor. The reviewers have opted to remain anonymous.

The reviewers have discussed their reviews with one another, and the Reviewing Editor has drafted this letter with what we consider essential revisions to help you prepare a revised submission. Most importantly, the authors need to show that Clstn3 is really the predominant Clstn3 expressed in Purkinje cells and that Clstn3 was specifically knocked out in Purkinje cells. These are the prerequisite conditions for the authors' conclusions as currently formulated.

Essential revisions:

A. Cell type specificity

1) According to the in situ hybridization data (Figure 1C), Clstn3 signals are detected in molecular and granular layers. I think that AAV-DJ using a U6 promoter to express gRNAs could infect various types of cells in the cerebellum. Since virus vectors were injected at various depths up to 2 mm, Clstn3 could be deleted in molecular-layer interneurons (MLI) that make inhibitory synapses onto Purkinje cells. Since Clstn3 could play roles in axonal transport, inhibitory synapses could have been impaired at least partly by presynaptic effects of Clstn3 in these cells. Clstn3 signals in granular layers, consisting of granule cells, Golgi cells and Lugaro cells, could also indirectly affect excitatory and inhibitory synapses in Purkinje cells. Thus, the authors need to ensure Purkinje-cell specific knockout of Clstn3 using specific promotors. Alternatively, the authors need to show that Clstn3 expression was unaffected in these neurons if the authors think that Clstn3 was specifically deleted in Purkinje cells by their AAV-DJ-U6 system.

2) It would have been far better to selectively target Purkinje cells, but that was not done. For that reason, the authors need to do a thorough job of dealing with all of the potential complications. The Allen brain atlas shows that Clstn2 and Clstn3 are both expressed by Purkinje cells and that Clstn3 is also expressed by stellate cells and basket cells. The authors used a viral approach that did not select for Purkinje cells. As a result, the effects they see on synaptic transmission could arise from the elimination of all cells in a region rather than the selective elimination of calsyntenin-3 in Purkinje cells. This changes the interpretation of many of the experimental results. For this reason, it is crucial that the carefully determine whether Clstn2 is present in Purkinje cells, and whether Clstn3 is present in stellate cells and basket cells.

3) An issue related to virus vectors is that although the authors used tdTomato as an expression marker of gRNAs (line 156), tdTomato driven by the CAG promoter does not necessarily ensure expression of gRNAs driven by the U6 promoter. Indeed, I feel that it is strange that AAV injected at various depths up to 2 mm did not reach the layer at all (Figure 2E). I suspect that the 20% decrease in VGAT signals in the granular layer (Figure 3D) may be caused by deletion of Clstn3 in granule cells, Golgi or Lugaro cells.

4) According to the in situ hybridization data (Figure 1C), Clstn2 is also expressed in Purkinje cells, weaking the authors' argument about the advantage of using these cells to prevent compensation by Clstns member. Thus, the authors need to show that the deletion of Clstn3 did not affect the amount and the localization of the Clstn2 in Purkinje cells. Immunoblot data shown in Figure S1A insufficient for this purpose.

5) There is no description about how the authors examined off-target effects of CRISPR/Cas9 (Figure S2). It is completely unclear how they collected AAV-infected cells and determined the genome sequence. If they could show AAV-infected cells are Purkinje cells here, it would be strengthen the authors' argument about the cell-type specific knockout. Rescue of knockout phenotypes by re-introduction of gRNA-insensitive Clstn3 will be more helpful to rule out the off-target effects.

B. Synapse numbers need to be assessed.

1) Although the spine density is a reliable proxy for excitatory synapses in the hippocampus, it is not the case for Purkinje cells. Unlike the description in line 267, spines are known to be formed cell autonomously in the absence of presynaptic parallel fibers in Purkinje cells (Sotelo, 1975). For example, the number of dendritic spines is unaffected even in cbln1- and grid2-null mice, which showed severe reduction in the number of parallel fiber synapses. I think that the authors need to use pre- and postsynaptic markers.

2) Counting the number of dendritic spines is not straightforward in Purkinje cells that extend numerous spines in all directions. I am not fully convinced by the morphological analyses of dendritic spines of Purkinje cells performed with Airyscan2. To address points 1) and 2), I would recommend using electron microscopic analyses, ideally with serial sections.

3) For inhibitory synapses, what the authors showed was the changes in presynaptic markers. If the authors want to discuss synapse formation/maintenance, I think they should use pre- and postsynaptic markers to estimate the number of inhibitory "synapses."

4) To strengthen the authors' conclusion about the specific role of Clstn3 in regulation of parallel-fiber synapses, they should show that Clstn3 is localized at parallel-fiber, but not climbing-fiber, synapses in wild-type Purkinje cells.

C. Issues associated with behavioral experiments.

1) Although the authors concluded that social interactions were not impaired in Clstn3 CRISPR KO mice (line 170), there is a tendency toward reduced sociability (Figure S3B, C). The authors need to increase the number of animals to reach a conclusion.

2) The behavioral data is not very compelling and is far from complete. It seems as if the rotarod assay was normal initially, and they just kept doing experiments until by trial there was a statistically significant difference. I think that this is of interest if it holds up, because it is a challenge to influence large parts of the cerebellum with AAV. I also think it is potentially very interesting, because the E/I balance seems to be strongly perturbed in these mice. Many other behavioral tests that are available to assess motor performance were not used. The number of experiments contributing to S3 is very small and it is not possible to conclude one way or anything regarding these behaviors. The number of experiments needs to be increased substantially or S3 should be eliminated.

[Editors' note: further revisions were suggested prior to acceptance, as described below.]

Thank you for resubmitting your work entitled "Calsyntenin-3, an atypical cadherin, suppresses inhibitory synapses but increases excitatory parallel-fiber synapses in cerebellum" for further consideration by *eLife*. Your revised article has been evaluated by Gary Westbrook (Senior Editor). The manuscript has been improved but there are some remaining issues that need to be addressed, as outlined below. We appreciate that these issue do not affect the main conclusions, but these issues not to be addressed in the text before we can render a final decision.

*Reviewer #2:*

I think the authors have thoroughly addressed comment A (Cell type specificity) of the Essential Revisions. Regarding comment B (Synapse numbers need to be assessed), there seems to be some misunderstanding. The point I wanted to make was that the spine density is NOT a reliable proxy for excitatory synapses in Purkinje cells. Indeed, in the review by Sudhof (2017) that the author cited, it is written as "Importantly, the loss of parallel-fiber synapses is NOT associated with a decrease in spine density." Thus, unlike in the hippocampus and cerebral cortex, the changes in the number of PF "synapses" are NOT reliably reflected by the number of Purkinje cell "spines." In other words, even if the number of dendritic spines is slightly increased, it is unclear whether they form synapses with PF terminals. The number of PF synapses can be counted easily by single-section EM analyses. However, considering that the major concern (comment A) has been thoroughly addressed by many additional experiments, I think authors could address comment B by rewording text. Similarly, the authors' responses to comment C (behavioral analyses) are OK.

*Reviewer #3:*

The authors have greatly improved the manuscript. They have done a good job of responding to most of the issues. In many cases their life would have been much easier had they simply taken a different experimental approach. For example, they could have done their manipulations selectively in Purkinje cells, but instead they had to do many controls to rule out alternative interpretations. In the end, they provide compelling data that supports their main conclusions. In so doing they have established that in cerebellar Purkinje cells, the acute knockout of calsyntenin-3 increases granule cell synapse density and decreases inhibitory synapse density. This is the first study showing that a particular synaptic adhesion molecule differentially regulates excitatory (increased) and inhibitory (decreased) synapses.

1. As stated in the initial review, the authors have a tendency to draw conclusions that go beyond their data. They responded these issues by stating that it was a standard in the field, based on a previous study that took this approach.

- Specifically, when asked about the 0-60 and >60 analysis they stated that it was standard in the field. While there may be some papers that adopt this approach, I do not think it is a justified approach. I do not agree that it can be used to discriminate between stellate cell and basket cell inputs. They should not use such a poorly justified approach simply because it has been used by others.

-The same approach was taken in response to the 1ms rise time being used to distinguish between parallel fiber and climbing fiber mEPSCs. They state "The claim that the rise time of 1 ms distinguishes between climbing- and parallel-fiber mEPSCs is not ours, but well documented in the literature (Yamasaki et al., 2006). We agree that it is likely only an approximation, but as an approximation we believe -as do others- that it is useful." They may believe it, but I think it is very weak and if they do use it they need point out how flimsy the evidence is for this. Finding a paper that uses this approach and turning into a standard for the field is not something I agree with.

2. They were asked for an explanation of how they determined the capacitance and membrane resistances, rather than simply providing the details of the methods, they gave a long and not very helpful explanation. Please describe the methods in detail. The issue of active conductances, including hyperpolarization-activated cation current, could certainly can come into play. The passive properties of the PCs are very important for the interpretation of the data, so detailed methods are required.

3. They were asked to strengthen the authors' conclusion about the specific role of Clstn3 in regulation of parallel-fiber synapses, they should show that Clstn3 is localized at parallel-fiber, but not climbing-fiber, synapses in wild-type Purkinje cells. They responded: As cited in the manuscript, Clstn3 has already been localized to parallel-fiber synapses in the cerebellum by immuno-EM in published papers. This does not address the issue of whether Clstn3 is absent from climbing fiber synapses.

[Editors' note: further revisions were suggested prior to acceptance, as described below.]

Thank you for resubmitting your work entitled "Calsyntenin-3, an atypical cadherin, suppresses inhibitory synapses but increases excitatory parallel-fiber synapses in cerebellum" for further consideration by *eLife*. Your revised article has been evaluated by Gary Westbrook (Senior Editor) and a Reviewing Editor.

The manuscript has been improved but there are some remaining issues that need to be addressed, as outlined below:

Essential revisions:

The editor and reviewers acknowledge that the authors made a real effort to address the previous reviews and that a large amount of work is included in the manuscript. However there are some remaining issues that we think need to be addressed as follows based on the re-reviews and the discussion between editors and reviewers:

1. The sparse lentivirus expression experiments are a great addition that satisfies many concerns for a subset of experiments. However, these additions raise serious concerns about the interpretation of all the experiments that were not studied using sparse labeling. The new results establish that the original experiments could be directly affecting MLIs and granule cells when many cells in a region are targeted. It must be assumed that direct effects on granule cells and MLIs contribute to any effects they describe, whether it be behavior or electrophysiology. The control experiments using sparse expression are only useful for those specific experiments that were performed. It is only valid to present data in which they have shown sparse and dense viral targeting show quantitatively the same result. Other data are uninterpretable at this point. The authors seem to have used the new experiments as a blanket control experiment to justify the interpretation of all experiments as being attributable to Clstn2 in Purkinje cells. This is drawing conclusions that go beyond what is supported by the data. The authors must be more careful in drawing conclusions from what is supported by the data.

2. Overall the behavioral experiments are weak. One basic issue is that a lack of effect could be attributed to not affecting all of the Purkinje cells that contribute to a behavior. The second is that the approach used for behavioral studies leaves open the possibility that off target effects could be responsible for the behavioral effects (as in point 1). There are good Cre lines for Purkinje cells. It would have been straightforward to selectively target Purkinje cells and that is what should have been done. The authors decided against using this approach and did not address the use of this approach in their response. Without selectively targeting Purkinje cells, behavioral disruptions are basically uninterpretable and the authors must acknowledge these limitations.

3. The authors imply that Clstn3 expression at higher levels in Purkinje cells than in other cells suggests that it means that Clstn3 is unlikely to be important in other cell types. This is not convincing logic as there certainly seems to be appreciable Clstn3 in other cell types. The high level of Clstn3 in granule cells and basket cells highlights the problem in interpreting many of their experiments. The authors need to adjust their conclusions in this regard, short of redesigning experiments that specifically target Purkinje cells.

4. In the response on page 6 the authors dismiss the prevalence of naked spines, which disregards a recent *eLife* paper (Miyzaki e al. *eLife* 2021) in which the elimination of granule cells and their associated presynaptic boutons, albeit in mature mice, left behind Purkinje cells covered with normal looking spines that lacked presynaptic boutons. Even if the spines are formed initially, they remain in the adults. Overall, the spine density is NOT a reliable proxy for excitatory synapses in Purkinje cells. Indeed, in the review by Sudhof (2017) that the author cited, it is written as "Importantly, the loss of parallel-fiber synapses is NOT associated with a decrease in spine density." Thus, unlike in the hippocampus and cerebral cortex, the changes in the number of PF "synapses" are NOT reliably reflected by the number of Purkinje cell "spines." In other words, even if the number of dendritic spines is slightly increased, it is unclear whether they form synapses with PF terminals. Thus, spine density is a flawed measure for excitatory synapses that should be acknowledged.

---

## [Author Response]

The reviewers have discussed their reviews with one another, and the Reviewing Editor has drafted this letter with what we consider essential revisions to help you prepare a revised submission. Most importantly, the authors need to show that Clstn3 is really the predominant Clstn3 expressed in Purkinje cells and that Clstn3 was specifically knocked out in Purkinje cells. These are the prerequisite conditions for the authors' conclusions as currently formulated.

In the revised paper, we have performed multiple experiments and data analyses to address this point. Specifically:

1) We have analyzed single-cell RNAseq data of the cerebellum which demonstrate that in the cerebellum, Purkinje cells uniquely express Clstn3, and that Clstn3 is not expressed at high levels in other cerebellar cells (new Figure 1—figure supplement 1A).

2) We have performed quantitative RT-PCR analyses of cytosol aspirated from Purkinje cells and other neurons with primers that were validated specifically for this purpose to demonstrate that Purkinje cells express Clstn3 at > 50-fold higher levels than Clstn1 or Clstn2, and that the levels of Clstn3 are much lower in other cerebellar cells (see figure below; data are in new Figure 1D, 1E, and Figure 1—figure supplement 1B-1E).

3) We analyzed Purkinje cells after the Clstn3 deletion using single-cell qRT-PCR with pipette-aspirate cytosol, and demonstrate that only Clstn3 but not other calsyntenins are suppressed by CRISPR-mediated deletion (new Figure 11—figure supplement 1).

Together, we believe that these results conclusively demonstrate that a) Purkinje cells express Clstn3 at much higher levels than other cerebellar cells, and b) other cerebellar cells do not express Clstn3 at similarly high levels.

Essential revisions:A. Cell type specificity1) According to the in situ hybridization data (Figure 1C), Clstn3 signals are detected in molecular and granular layers. I think that AAV-DJ using a U6 promoter to express gRNAs could infect various types of cells in the cerebellum. Since virus vectors were injected at various depths up to 2 mm, Clstn3 could be deleted in molecular-layer interneurons (MLI) that make inhibitory synapses onto Purkinje cells. Since Clstn3 could play roles in axonal transport, inhibitory synapses could have been impaired at least partly by presynaptic effects of Clstn3 in these cells. Clstn3 signals in granular layers, consisting of granule cells, Golgi cells and Lugaro cells, could also indirectly affect excitatory and inhibitory synapses in Purkinje cells. Thus, the authors need to ensure Purkinje-cell specific knockout of Clstn3 using specific promotors. Alternatively, the authors need to show that Clstn3 expression was unaffected in these neurons if the authors think that Clstn3 was specifically deleted in Purkinje cells by their AAV-DJ-U6 system.

We agree that in our experiments, the AAVs we use likely infect other neurons in addition to Purkinje cells, and that in these neurons Clstn3 expression will also be suppressed. Thus, we concur that it is conceivable that the phenotype we observed could also be due to a suppression of Clstn3 in other cerebellar neurons, and that our data did not conclusively rule out the possibility of a presynaptic function of Clstn3 in excitatory granule cells or in inhibitory basket/stellate cells in the cerebellum. To address this concern, we have performed two sets of new experiments:

1) We have used sparse lentiviral infections to delete Clstn3 only in a few isolated Purkinje cells of the entire cerebellum of an animal. In these experiments, a single Purkinje cell is infected in a broad section of the cerebellar cortex (see panel B, left image). We then demonstrate that with these sparse Clstn3 deletions, the Purkinje cells exhibit the same phenotype as with the broad deletions of the entire cerebellar Clstn3 using AAVs. Thus, Clstn3 acts cell-autonomously in Purkinje cells (new Figure 11). Note that in these sparse deletions of Clstn3, we observe the identical and intriguing increase in parallel-fiber EPSCs and suppression of basket/stellate cell IPSCs as in AAV-mediated Clstn3 deletions. This experiment is better than the suggested experiment with a Purkinje cell-specific promoter because in viral vectors, cell type-specific promoters exhibit limited specificity, and only transgenic methods are suitable to achieve true cell type specificity.

2) In parallel, we have analyzed the effect of the sparse deletion of Clstn3 in Purkinje cells on the expression of Clstn1, Clstn2, and Clstn3 in Purkinje cells and in surrounding neurons (new Figure 11—figure supplement 1; see image below). Because the lentiviral infections were so sparse, these were very difficult experiments with only a few cells per animal. Nevertheless, the data show that the sparse deletion of Clstn3 in Purkinje cells only lowers Clstn3, but not Clstn1 or Clstn2, levels in Purkinje cells, and that it only lowers Clstn3 levels in Purkinje cells but not in surrounding basket or granule cells (see Figure 11; new Figure 11—figure supplement 1). We believe that this experiment conclusively demonstrates that it is truly the postsynaptic deletion of Clstn3 that causes the robust phenotype we observe, which is even more impactful considering that the KO is unlikely to be 100% effective since it is not a conditional deletion.

3) Finally, we would like to note that given the relatively specific expression of Clstn3 in Purkinje cells compared to other cerebellar neurons, it is unlikely that the Clstn3 deletion operates in these other neurons, although we do agree that without the new experiments we just described above, we could not have ruled out this unlikely possibility.

We hope that these new data will be persuasive in showing that it is truly the deletion of postsynaptic Clstn3 that produces the strong phenotype we observe.

2) It would have been far better to selectively target Purkinje cells, but that was not done. For that reason, the authors need to do a thorough job of dealing with all of the potential complications. The Allen brain atlas shows that Clstn2 and Clstn3 are both expressed by Purkinje cells and that Clstn3 is also expressed by stellate cells and basket cells. The authors used a viral approach that did not select for Purkinje cells. As a result, the effects they see on synaptic transmission could arise from the elimination of all cells in a region rather than the selective elimination of calsyntenin-3 in Purkinje cells. This changes the interpretation of many of the experimental results. For this reason, it is crucial that the carefully determine whether Clstn2 is present in Purkinje cells, and whether Clstn3 is present in stellate cells and basket cells.

This are essentially the same concerns as expressed in the previous two comments, namely the fact that Clstn3 is also expressed in other neurons besides Purkinje cells, that Purkinje cells could also express other calsyntenins, and that the AAV infection of the cerebellar cortex could also have deleted Clstn3 in other neurons. These concerns were addressed in the new experiments and analyses described above.

3) An issue related to virus vectors is that although the authors used tdTomato as an expression marker of gRNAs (line 156), tdTomato driven by the CAG promoter does not necessarily ensure expression of gRNAs driven by the U6 promoter. Indeed, I feel that it is strange that AAV injected at various depths up to 2 mm did not reach the layer at all (Figure 2E). I suspect that the 20% decrease in VGAT signals in the granular layer (Figure 3D) may be caused by deletion of Clstn3 in granule cells, Golgi or Lugaro cells.

This comment voices three concerns that we have addressed as follows:

a) The concern that tdTomato expression by a virus via the CAG promoter does not guarantee expression of the gRNA via the U6 promoter from the same virus. We completely agree, which is why we carefully demonstrate that Clstn3 mRNAs are severely suppressed by the gRNA as shown by multiple types of measurements (see Figure 2 and Figure 11—figure supplement 1 in the revised paper).

b) The concern that it is strange that AAVs injected into the cerebellar cortex apparently did not reach the upper layers of the cerebellar cortex. The original Figure 2E (now Figure 2F in the revised paper) did indeed appear to suggest that the upper layers of the cerebellar cortex are not infected, but we have now added a new supplementary figure (Figure 2—figure supplement 2) to show that they are. We think that part of the confusion here derives from the way we designed our experiments, which use tdTomato expression from the virus as an indicator, NOT Cre-expression by the virus in a Cre-dependent tdTomato indicator mouse. In the latter case, very low levels of Cre expression are sufficient to elicit a tdTomato signal which is uniform across the infected tissue. This can be misleading. In our case of virally expressed tdTomato, the levels of tdTomato reflect the viral titer, and it is to be expected that these levels are highest closer to the injection sites, and lower at more distant sites.

c) The concern that the loss of inhibitory synapses via the cerebellar CRISPR-mediated deletion of Clstn3 is due to a suppression of Clstn3 expression in inhibitory neurons where Clstn3 might have a presynaptic function instead of the suppression of Clstn3 expression in Purkinje cells. This valid concern is the same concern voiced in previous comments, and was addressed above with the sparse deletion of Clstn3 only in Purkinje cells and the demonstration that Clstn3 is expressed at high levels only in Purkinje cells.

4) According to the in situ hybridization data (Figure 1C), Clstn2 is also expressed in Purkinje cells, weaking the authors' argument about the advantage of using these cells to prevent compensation by Clstns member. Thus, the authors need to show that the deletion of Clstn3 did not affect the amount and the localization of the Clstn2 in Purkinje cells. Immunoblot data shown in Figure S1A insufficient for this purpose.

Agreed – we have addressed this concern with the following new data:

a) Single-cell qRT-PCR analyses of wild-type mice demonstrate that Clstn3 is expressed at >25-fold higher levels in Purkinje cells than Clstn1 or Clstn2 (new Figure 1; see data shown above on p. 1).

b) Single-cell qRT-PCR analyses of sparsely infected Purkinje cells show that the CRISPR deletion affects only Clstn3 levels but not Clstn1 or Clstn2 (new Figure 11—figure supplement 1).

c) Analyses of single-cell RNAseq data deposited by the McCarroll lab show that, consistent with our qRT-PCR studies, Purkinje cells express only low amounts of Clstn1 or Clstn2, which are, however, expressed at higher levels in other cerebellar cells (new Figure 1—figure supplement 1A).

5) There is no description about how the authors examined off-target effects of CRISPR/Cas9 (Figure S2). It is completely unclear how they collected AAV-infected cells and determined the genome sequence. If they could show AAV-infected cells are Purkinje cells here, it would be strengthen the authors' argument about the cell-type specific knockout. Rescue of knockout phenotypes by re-introduction of gRNA-insensitive Clstn3 will be more helpful to rule out the off-target effects.

We apologize for the lack of clarity and have improved this in the revised paper. Please note that we analyzed the off-target effects of our CRISPR manipulations using bioinformatics and genomic PCRs, but did not perform whole-genome sequencing to confirm these analyses (which would be another large project). We concur that rescue experiments would be useful, but they exceed the scope of the current study, which is already a massive project spanning many years of work and now amounting to 11 data figures. Please also note that we have now addressed the cell-type specificity with the sparse lentiviral infection as described above.

B. Synapse numbers need to be assessed.1) Although the spine density is a reliable proxy for excitatory synapses in the hippocampus, it is not the case for Purkinje cells. Unlike the description in line 267, spines are known to be formed cell autonomously in the absence of presynaptic parallel fibers in Purkinje cells (Sotelo, 1975). For example, the number of dendritic spines is unaffected even in cbln1- and grid2-null mice, which showed severe reduction in the number of parallel fiber synapses. I think that the authors need to use pre- and postsynaptic markers.

The reviewer expresses a view of the ‘naked spines’ in GluD2 and Cbln1 mutant mice that is still written in many textbooks, but that is simply incorrect. Claudio Sotelo was a giant in the field of cerebellar anatomy, but much has been done since. Spines are NOT known to be formed cell-autonomously – the famous ‘naked spines’ in the Cbln1 and Grid2 mutant mice arise because synapses are initially formed normally, but then degenerate (please see the discussion in Sudhof (2017) for details). What happens, simply, is that defective synaptic transmission leads to a degradation of nerve terminals first, and then spines. Spine numbers are not normal but decreased in these mutants, and the prevalence of ‘naked spines’ is rare. An extensive and beautiful literature, primarily from leading Japanese labs in studies published from 1990 to 2010, documents this (e.g., see Hirai, et al., 2005, Uemura, et al., 2010).

2) Counting the number of dendritic spines is not straightforward in Purkinje cells that extend numerous spines in all directions. I am not fully convinced by the morphological analyses of dendritic spines of Purkinje cells performed with Airyscan2. To address points 1) and 2), I would recommend using electron microscopic analyses, ideally with serial sections.

We agree that 3D reconstructions using serial EM sections would be great and appreciate the recommendation to perform such analyses for wild type and mutant cerebellum. However, this has not yet been achieved for even the wild-type cerebellum – it would basically represent a >5-year project given the large size of the Purkinje cell dendritic arbors and the enormous degree of connectivity of parallel fiber synapses. BUT: We did perform full 3D reconstructions by confocal microscopy, using advanced microscopes and highly respected software! Although these reconstructions took months to accomplish for a limited number of cells, the results are unequivocal and represent the state-of-the-art in this field. Thus, we did not just count spines in sections, but performed actual reconstructions that enable counting numerous spines in all directions.

3) For inhibitory synapses, what the authors showed was the changes in presynaptic markers. If the authors want to discuss synapse formation/maintenance, I think they should use pre- and postsynaptic markers to estimate the number of inhibitory "synapses."

In new experiments we have measured the density of inhibitory synapses using antibodies to GABA-Aa1 receptors in matched control and Clstn3-deficient cerebellar cortex (new Figure 4E-4H, also the left figure). The results confirm the previous conclusions, which are also supported by the measurements of mIPSCs.

4) To strengthen the authors' conclusion about the specific role of Clstn3 in regulation of parallel-fiber synapses, they should show that Clstn3 is localized at parallel-fiber, but not climbing-fiber, synapses in wild-type Purkinje cells.

As cited in the manuscript, Clstn3 has already been localized to parallel-fiber synapses in the cerebellum by immuno-EM in published papers.

C. Issues associated with behavioral experiments.1) Although the authors concluded that social interactions were not impaired in Clstn3 CRISPR KO mice (line 170), there is a tendency toward reduced sociability (Figure S3B, C). The authors need to increase the number of animals to reach a conclusion.

As recommended, we have greatly expanded the behavioral analyses of the mice in which cerebellar Clstn3 was deleted, including more exhaustive analysis of social interactions. The new data are now shown in Figure 3.

2) The behavioral data is not very compelling and is far from complete. It seems as if the rotarod assay was normal initially, and they just kept doing experiments until by trial there was a statistically significant difference. I think that this is of interest if it holds up, because it is a challenge to influence large parts of the cerebellum with AAV. I also think it is potentially very interesting, because the E/I balance seems to be strongly perturbed in these mice. Many other behavioral tests that are available to assess motor performance were not used. The number of experiments contributing to S3 is very small and it is not possible to conclude one way or anything regarding these behaviors. The number of experiments needs to be increased substantially or S3 should be eliminated.

The protocol we use for the rotarod assay is a standard method using an escalating challenge. This protocol is not an afterthought, but was described in detail in our Rothwell et al., 2014 paper as cited. The protocol was developed to measure motor learning, which is clearly impaired in our mice. We have now expanded the behavioral analysis and show this analysis in the new Figure 3. The bottom line is that a discrete impairment of motor behaviors can be reproducibly observed with no changes in other behavioral parameters.

[Editors' note: further revisions were suggested prior to acceptance, as described below.]

Reviewer #2:I think the authors have thoroughly addressed comment A (Cell type specificity) of the Essential Revisions. Regarding comment B (Synapse numbers need to be assessed), there seems to be some misunderstanding. The point I wanted to make was that the spine density is NOT a reliable proxy for excitatory synapses in Purkinje cells. Indeed, in the review by Sudhof (2017) that the author cited, it is written as "Importantly, the loss of parallel-fiber synapses is NOT associated with a decrease in spine density." Thus, unlike in the hippocampus and cerebral cortex, the changes in the number of PF "synapses" are NOT reliably reflected by the number of Purkinje cell "spines." In other words, even if the number of dendritic spines is slightly increased, it is unclear whether they form synapses with PF terminals. The number of PF synapses can be counted easily by single-section EM analyses. However, considering that the major concern (comment A) has been thoroughly addressed by many additional experiments, I think authors could address comment B by rewording text. Similarly, the authors' responses to comment C (behavioral analyses) are OK.

We agree that spine numbers are not a completely reliable proxy for synapse numbers, although in most cases quite accurate. However, but not always a reliable one since there are situations in which pre- and postsynaptic specializations diverge. However, since we also measured synapse numbers by immunocytochemistry (which has its own acknowledged limitations), we believe our conclusions are justified. We have now made this point clearer in the paper.

Reviewer #3:The authors have greatly improved the manuscript. They have done a good job of responding to most of the issues. In many cases their life would have been much easier had they simply taken a different experimental approach. For example, they could have done their manipulations selectively in Purkinje cells, but instead they had to do many controls to rule out alternative interpretations. In the end, they provide compelling data that supports their main conclusions. In so doing they have established that in cerebellar Purkinje cells, the acute knockout of calsyntenin-3 increases granule cell synapse density and decreases inhibitory synapse density. This is the first study showing that a particular synaptic adhesion molecule differentially regulates excitatory (increased) and inhibitory (decreased) synapses.

We appreciate the positive comments.

1. As stated in the initial review, the authors have a tendency to draw conclusions that go beyond their data. They responded these issues by stating that it was a standard in the field, based on a previous study that took this approach.- Specifically, when asked about the 0-60 and >60 analysis they stated that it was standard in the field. While there may be some papers that adopt this approach, I do not think it is a justified approach. I do not agree that it can be used to discriminate between stellate cell and basket cell inputs. They should not use such a poorly justified approach simply because it has been used by others.

We realize that the reviewer is deeply concerned about the approach of classifying mIPSCs via their amplitude, and that he/she feels that this approach is wrong even though it has been used by others. To accommodate this concern, we have now reworded the description of this analysis in the paper to exclude the claim that the mIPSCs with different amplitudes are differentially derived from basket and stellate cells, and only use this classification to show that the phenotype we observe persists in different subsets of mIPSCs identified by their differential amplitudes. We mention that some people in the field believe that these different amplitudes correspond to different neuronal origins, but make no claim that this is necessarily a fact. We hope the reviewer will agree with these very cautious descriptions.

-The same approach was taken in response to the 1ms rise time being used to distinguish between parallel fiber and climbing fiber mEPSCs. They state "The claim that the rise time of 1 ms distinguishes between climbing- and parallel-fiber mEPSCs is not ours, but well documented in the literature (Yamasaki et al., 2006). We agree that it is likely only an approximation, but as an approximation we believe -as do others- that it is useful." They may believe it, but I think it is very weak and if they do use it they need point out how flimsy the evidence is for this. Finding a paper that uses this approach and turning into a standard for the field is not something I agree with.

We also agree that using the criterion of the mEPSC kinetics is an imperfect approach to distinguishing between parallel- and climbing-fiber synapses. However, based on classical studies in the field, such as the Magee and Cook 2008 paper that beautifully documents the relation of mEPSC kinetics to dendritic distance, we do feel this criterion has some validity. To assuage the reviewer’s concerns and follow her/his recommendation, we have now emphasized even more strongly in the paper that this classification of mEPSCs is only an approximation, not a definitive measurement.

2. They were asked for an explanation of how they determined the capacitance and membrane resistances, rather than simply providing the details of the methods, they gave a long and not very helpful explanation. Please describe the methods in detail. The issue of active conductances, including hyperpolarization-activated cation current, could certainly can come into play. The passive properties of the PCs are very important for the interpretation of the data, so detailed methods are required.

We have now added a detailed methods description in the revised manuscript.

3. They were asked to strengthen the authors' conclusion about the specific role of Clstn3 in regulation of parallel-fiber synapses, they should show that Clstn3 is localized at parallel-fiber, but not climbing-fiber, synapses in wild-type Purkinje cells. They responded: As cited in the manuscript, Clstn3 has already been localized to parallel-fiber synapses in the cerebellum by immuno-EM in published papers. This does not address the issue of whether Clstn3 is absent from climbing fiber synapses.

As cited in the manuscript, previous studies localized Clstn3 to parallel-fiber but not climbing-fiber synapses. However, this finding does not rule out a low concentration of Clstn3 in climbing-fiber synapses. We cannot exclude the presence of some Clstn3 molecules from climbing-fiber synapses, nor can anyone else for that matter, because any immunocytochemical or imaging technique will never be sensitive enough to rule out levels of a synaptic molecule in a synapse if it present in the same neurons at other synapses at much higher levels. We strongly believe that the lack of an effect of the Clstn3 KO on climbing-fiber synaptic transmission shows that Clstn3 has no major function at these synapses. We believe this is conclusive, and that most people in the field would agree, no matter whether or not low concentrations of Clstn3 might be present in climbing-fiber synapses. We have now added a sentence to that effect in the paper.

[Editors' note: further revisions were suggested prior to acceptance, as described below.]

Essential revisions:The editor and reviewers acknowledge that the authors made a real effort to address the previous reviews and that a large amount of work is included in the manuscript. However there are some remaining issues that we think need to be addressed as follows based on the re-reviews and the discussion between editors and reviewers:1. The sparse lentivirus expression experiments are a great addition that satisfies many concerns for a subset of experiments. However, these additions raise serious concerns about the interpretation of all the experiments that were not studied using sparse labeling. The new results establish that the original experiments could be directly affecting MLIs and granule cells when many cells in a region are targeted. It must be assumed that direct effects on granule cells and MLIs contribute to any effects they describe, whether it be behavior or electrophysiology. The control experiments using sparse expression are only useful for those specific experiments that were performed. It is only valid to present data in which they have shown sparse and dense viral targeting show quantitatively the same result. Other data are uninterpretable at this point. The authors seem to have used the new experiments as a blanket control experiment to justify the interpretation of all experiments as being attributable to Clstn2 in Purkinje cells. This is drawing conclusions that go beyond what is supported by the data. The authors must be more careful in drawing conclusions from what is supported by the data.

We regret to say that we disagree with this comment, which may have been due to our inability to clearly communicate the logic of our experiments. The reviewer(s) offers the fundamental argument that most of our experiments are uninterpretable because they were performed by manipulations that are not cell type-specific. If this argument was correct, the vast majority of the literature on synaptic mechanisms on the cerebellum would have to be dismissed, including Yuzaki’s classical studies on the *Cbln1* KO, de Zeuw’s pioneering work on GluA1, and Mishina’s discoveries of GluD2. All of this with one sweep would be uninterpretable because none of these studies used cell-type specific manipulations. These authors argued previously, we think convincingly, that the context of their experiments allow definitive interpretable conclusions, and we hope that the same applies to our experiments.

To recap, our fundamental observations are that (a) the acute Calsyntenin-3 deletion using in vivo CRISPR introduced by AAVs in the cerebellar cortex caused an increase in parallel-fiber synapse numbers and a decrease in GABAergic synapses on Purkinje cells, as documented by multiple electrophysiological measurements, staining for a panoply of synaptic markers, and reconstruction of Purkinje cells; (b) Calsyntenin-3 is expressed at much higher levels in Purkinje cells than in other neurons or glia of the cerebellum, such that granule cells express >10-fold lower levels and basket cells >5-fold lower levels of Calsyntenin-3 than Purkinje cells; (c) the AAV stain that we use infects Purkinje cells preferentially; and (d) the sparse Calsyntenin-3 deletion in Purkinje cells using lentiviruses causes the same electrophysiological phenotype as the deletion of Calsyntenin-3 in many Purkinje cells by AAVs.

Based on these combined findings, we conclude that Calsyntenin-3 functions both to promote and to restrict synapse formation, and that Calsyntenin-3 acts in Purkinje cells. We did not rule out a contribution of Calsyntenin-3 to the function of other cerebellar neurons. We would like to argue that the findings speak for themselves: postsynaptic Calsyntenin-3 in Purkinje cells acts as a major regulator of inhibitory basket/stellate-cell and or parallel-fiber synapses. Compared for example to the *Cbln1* deletion, the effect sizes we see are large, and surely Calsyntenin-3 emerges here as a significant contributor to shaping cerebellar circuits.

We have modified the text to make sure that the argument we make is clearly articulated, and hope this will now be acceptable to the scientific community and to the reviewers.

2. Overall the behavioral experiments are weak. One basic issue is that a lack of effect could be attributed to not affecting all of the Purkinje cells that contribute to a behavior. The second is that the approach used for behavioral studies leaves open the possibility that off target effects could be responsible for the behavioral effects (as in point 1). There are good Cre lines for Purkinje cells. It would have been straightforward to selectively target Purkinje cells and that is what should have been done. The authors decided against using this approach and did not address the use of this approach in their response. Without selectively targeting Purkinje cells, behavioral disruptions are basically uninterpretable and the authors must acknowledge these limitations.

This comment is similar to the first comment, and again may be partly due to our inability to communicate facts clearly, for which we apologize. The comment concludes that “*Overall the behavioral experiments are weak … Without selectively targeting Purkinje cells, behavioral disruptions are basically uninterpretable and the authors must acknowledge these limitations*”. We agree that the behavioral experiments are not extensive, but our paper is not intended as a major behavioral study, and the behavioral data occupy only one of 11 main and 8 supplementary figures. Nevertheless, the behavioral phenotype we observe is robust and consistent with a cerebellar dysfunction. It is conceivable that not all Purkinje cells were infected by our in vivo manipulations, but the presence of a major phenotype appears to us to be sufficient to argue that the physiological changes we observe are behaviorally relevant, which is the purpose of the behavioral experiments. The fact that the KO of a gene that is expressed primarily in Purkinje cells using a viral CRISPR manipulation that primarily affects Purkinje cells causes a major phenotype means to us that this gene is important for cerebellar functions, and that its effects are likely mediated by Purkinje cells.

We have modified the text to make sure that all readers understand that our study uses behavioral experiments as a way to document that the Calsyntenin-3 deletion in the cerebellum produces a biologically significant phenotype, and that we do not present an exhaustive behavioral analysis, although that might be apparent from the data.

3. The authors imply that Clstn3 expression at higher levels in Purkinje cells than in other cells suggests that it means that Clstn3 is unlikely to be important in other cell types. This is not convincing logic as there certainly seems to be appreciable Clstn3 in other cell types. The high level of Clstn3 in granule cells and basket cells highlights the problem in interpreting many of their experiments. The authors need to adjust their conclusions in this regard, short of redesigning experiments that specifically target Purkinje cells.

Again, we fear that we were not clear in our presentation since the reviewer states that there are high levels of Calsyntenin-3 in granule and basket cells, which is not what we show or what others have found. Our results and published in situ and RNAseq data show that Calsyntenin-3 expression in granule and basket cells is very low – in our in situ experiments, we cannot even detect Calsyntenin-3 in granule cells, only sensitive RT-PCR assays enable that.

Moreover, we have now made sure in the paper that we make no claims that Calsyntenin-3 is unlikely to be important in other cell types. Since we did not study other cerebellar cell types, we don’t know. Our conclusion is that Calsyntenin-3 in Purkinje cells regulates the formation/maintenance of parallel-fiber and basket/stellate cell synapses on the Purkinje cells. We do believe this conclusion is justified since we show (i) that Calsyntenin-3 is primarily expressed in Purkinje and cells, (ii) that a global viral manipulation that primarily if not exclusively affects Purkinje cells causes a large phenotype as assessed by a broad number of assays, and (iii) that a sparse manipulation of the same gene only in Purkinje cells replicates the electrophysiological phenotype of the global Purkinje cell deletion. We have now re-formulated our text to reflect this more clearly.

4. In the response on page 6 the authors dismiss the prevalence of naked spines, which disregards a recent eLife paper (Miyzaki e al. eLife 2021) in which the elimination of granule cells and their associated presynaptic boutons, albeit in mature mice, left behind Purkinje cells covered with normal looking spines that lacked presynaptic boutons. Even if the spines are formed initially, they remain in the adults. Overall, the spine density is NOT a reliable proxy for excitatory synapses in Purkinje cells. Indeed, in the review by Sudhof (2017) that the author cited, it is written as "Importantly, the loss of parallel-fiber synapses is NOT associated with a decrease in spine density." Thus, unlike in the hippocampus and cerebral cortex, the changes in the number of PF "synapses" are NOT reliably reflected by the number of Purkinje cell "spines." In other words, even if the number of dendritic spines is slightly increased, it is unclear whether they form synapses with PF terminals. Thus, spine density is a flawed measure for excitatory synapses that should be acknowledged.

We also apologize if we created the impression that we dismissed the possibility of naked spines, which was not our intention, and have now mentioned this possibility extensively and multiple times throughout the paper. But we have to confess that we do not quite understand the reviewers’ criticism here since the previous version already mentioned naked spines and explicitly said that they might contribute to the spine density, and that measurements of the spine density are only interpretable in the context of multiple approaches that probe synapse density.

In our experiments, we use the number of spines as one of 6 measurements to assess the density of excitatory parallel-fiber synapses. These measurements include in addition to spine density measurements: quantitative immunocytochemistry for vGluT1, GluA1, GluA2, mEPSCs, and evoked parallel-fiber EPSCs. It seems to us that, with a complete concordance of all six measurements, we are on firm grounds to conclude that parallel-fiber synapse numbers are increased. We agree that the increase in spines (which the reviewer calls ‘*slight’* but is the biggest ever observed as far as we could tell) by itself does not prove that there are more parallel-fiber synapses, but we do not claim this – we just use it as one of many measurements. It should also be noted that the naked spines do represent remnants of previous synapses – so they do reflect synapses that once existed – and that no condition every induced an increase in naked spines. We have modified the text to make sure that no reader feels we dismiss naked spines, and that all readers understand the multiple parameters we measured to assess parallel-fiber synapse numbers.